# Continuous-Time Piecewise-Linear Recurrent Neural Networks

**Alena Brändle** [* 1 2 3]   **Lukas Eisenmann** [* 1 2]   **Florian Götz** [1 4]   **Daniel Durstewitz** [1 2 4 5]

## Abstract

In dynamical systems reconstruction (DSR) we aim to recover the dynamical system (DS) underlying observed time series. Specifically, we aim to learn a generative surrogate model which approximates the underlying, data-generating DS, and recreates its long-term properties ('climate statistics'). In scientific and medical areas, in particular, these models need to be mechanistically tractable – through their mathematical analysis we would like to obtain insight into the recovered system's workings. Piecewise-linear (PL), ReLU-based RNNs (PLRNNs) have a strong track-record in this regard, representing SOTA DSR models while allowing mathematical insight by virtue of their PL design. However, all current PLRNN variants are *discrete-time maps*. This is in disaccord with the assumed continuous-time nature of most physical and biological processes, and makes it hard to accommodate data arriving at *irregular* temporal intervals. Neural ODEs are one solution, but they do not reach the DSR performance of PLRNNs and often lack their tractability. Here we develop theory for *continuous-time* PLRNNs (cPLRNNs): We present a novel algorithm for training and simulating such models, bypassing numerical integration by efficiently exploiting their PL structure. We further demonstrate how important topological objects like equilibria or limit cycles can be determined semi-analytically in trained models. We compare cPLRNNs to both their discrete-time cousins as well as Neural ODEs on DSR benchmarks, including systems with discontinuities which come with hard thresholds.

---

[1]Dept. of Theoretical Neuroscience, Central Institute of Mental Health (CIMH), Mannheim, Germany [2]Faculty of Physics and Astronomy, Heidelberg University [3]Hector Institute for Artificial Intelligence in Psychiatry, CIMH [4]Faculty of Mathematics and Computer Science, Heidelberg University [5]Interdisciplinary Center for Scientific Computing, Heidelberg University. Correspondence to: Alena Brändle <alena.braendle@zi-mannheim.de>, Daniel Durstewitz <daniel.durstewitz@zi-mannheim.de>.

*Proceedings of the 43rd International Conference on Machine Learning*, Seoul, South Korea. PMLR 306, 2026. Copyright 2026 by the author(s).

## 1. Introduction

Scientific theories for explaining and predicting empirical phenomena are most commonly formulated in terms of systems of differential equations, aka dynamical systems (DS). While traditionally this meant hand-crafting mathematical models, in the last ~5 years or so deep learning for automatically inferring dynamical models from time series data has become increasingly popular, also termed dynamical systems reconstruction (DSR) (Brunton et al., 2022; Durstewitz et al., 2023; Gilpin, 2024). To qualify as proper surrogate models for the underlying dynamical process, DSR models must fulfill certain criteria. Most importantly, they must be able to reproduce the long-term properties (ergodic statistics) of an observed DS (Göring et al., 2024). To be useful in scientific or medical contexts, they should also be interpretable and mathematically tractable, such that they can be analyzed to gain insight into dynamical mechanisms.

State of the art (SOTA) models which fulfill both criteria are piecewise-linear (PL) systems, like PL recurrent neural networks (PLRNNs; Brenner et al. (2022; 2024b); Hess et al. (2023); Pals et al. (2024)) or recursive switching linear DS (rSLDS; Linderman et al. (2016; 2017)). These models are tractable by their PL design (Monfared & Durstewitz, 2020; Eisenmann et al., 2023; 2026), properties for which they have a celebrated tradition in both engineering (Bemporad et al., 2000; Rantzer & Johansson, 2000; Carmona et al., 2002; Juloski et al., 2005; Stanculescu et al., 2014) and the mathematics of DS (Alligood et al., 1996; Avrutin et al., 2019; Simpson, 2025; Coombes et al., 2024). However, all these models are *discrete time recursive maps* and thus require binning of the time axis – they cannot naturally deal with data spaced across irregular temporal intervals, although these are quite common in various scientific disciplines like climate research or medical settings. Moreover, while theoretically discrete-time methods are supposed to approximate the flow (solution) operator of the underlying DS, practically this is not always given and may need additional regularization criteria to enforce the flow's semigroup properties (Li et al., 2022). This is not only an issue for training if observations are made at arbitrary time points, but also for inference as we would like to be able to inter- and extrapolate a system's state at arbitrary times. A natural solution is to formulate the model right away in terms of differential equations rather than maps, as in Neural ODEs

(Chen et al., 2018; Alvarez et al., 2020) or physics-informed neural networks (Raissi et al., 2019). However, these lack the appealing mathematical accessibility of PLRNNs, and also cannot compete with them performance-wise (Brenner et al., 2022; Hess et al., 2023).

To address this, here we introduce a continuous-time version of PLRNNs. In particular, we show how to harvest its PL structure for model training, simulation, and analysis. Our specific contributions are:

1. We introduce a *novel algorithm for solving PL ODE systems* exploiting the fact that *within* each linear subregion of the system's state space we have an analytical expression for the dynamics. Thus, instead of numerically integrating the system (as in Neural ODEs), we semi-analytically determine the switching times at which the trajectory crosses the boundary into a new linear subregion. This gives rise to a much more efficient and precise procedure, as we do not require a numerical solver which relies on determining optimal integration step sizes to meet a preset error criterion.

2. We demonstrate how the theory of PL continuous-time DS (Coombes et al., 2024) can be harvested *to determine important topological properties of the inferred DS, like its equilibria and limit cycles*. This makes important DS features semi-analytically tractable, enabling deeper mathematical insight into the system's behavior than feasible with previous methods.

We believe these are important steps for establishing data-inferred DS models as general scientific analysis and theory-building tools.

## 2. Related Work

**Dynamical systems reconstruction (DSR)**   In DSR we aim to infer a generative surrogate model from time series observation which reproduces the underlying system's long-term behavior (Durstewitz et al., 2023). Many directions toward this goal have been tested in past years, relying on models defined in terms of function libraries (Champion et al., 2019), RNNs including reservoir computers (Pathak et al., 2017; Platt et al., 2022; 2023), LSTMs (Vlachas et al., 2018), and PLRNNs (Durstewitz, 2017; Brenner et al., 2022; 2024a), or neural ODEs (Chen et al., 2018; Alvarez et al., 2020). One key aspect is the training process itself which needs to ensure that ergodic properties are captured, and various control-theoretically motivated training algorithms (Mikhaeil et al., 2022; Hess et al., 2023) or special loss criteria (Platt et al., 2022; 2023) have been suggested to accomplish this. Recent work dealt with multimodal data integration for the purpose of DSR, steps towards DSR foundation models (Brenner et al., 2024b; Hemmer & Durste-

witz, 2026), and the topic of out-of-domain generalization in DSR (Göring et al., 2024).

**Continuous-time RNNs**   Continuous-time models at some level seem the more natural choice for DSR, but currently face issues with mathematical tractability and still lag behind in DSR performance (Brenner et al., 2022; Hess et al., 2023). Early work on continuous-time RNNs dates back to the Wilson–Cowan equations (Wilson & Cowan, 1972), which describe neural population dynamics via coupled ODEs. Pearlmutter (1995) later generalized backpropagation to continuous-time RNNs, enabling learning in differentiable DS. Modern approaches parameterize the vector field with deep networks, as in Neural ODEs (Chen et al., 2018) which invoke the adjoint sensitivity method to efficiently compute gradients.

Numerous extensions to the basic approach have been advanced in subsequent years. Augmented Neural ODEs (Dupont et al., 2019) expand the state space to enhance expressivity and mitigate topological constraints, as well as improve and speed up model training. Latent Neural ODEs (Rubanova et al., 2019) develop the basic approach more specifically for irregularly sampled time series data. Neural Controlled Differential Equations (Kidger et al., 2020) generalize Neural ODEs to allow for continuous control signals (rendering the system strictly non-autonomous) and for naturally handling partial observations. Hamiltonian Neural Networks (Greydanus et al., 2019) explicitly incorporate an Hamiltonian formulation into the loss to learn physical systems with conservation laws, an approach later extended by the Symplectic ODE-Net (Zhong et al., 2020) which includes external forcing and control into the model. Other advancements of the basic approach allow for event functions (Zhong et al., 2020) or explicitly model second-order terms (Norcliffe et al., 2020). Neural Stochastic Differential Equations (SDEs) (Tzen & Raginsky, 2019; Li et al., 2020), finally, introduce stochasticity into the latent process as in SDEs by separately modeling the drift and diffusion terms.

**Piecewise linear systems**   Since in DSR we are not merely interested in prediction, but in gaining insight into the system dynamics which gave rise to the observed process, mathematical tractability is an important criterion. PL DS partition the state space into subregions with linear dynamics, allowing for a complete analytical description *within* each subregion. These properties have made PL maps one of the most intensely studied areas in engineering (Bemporad et al., 2000; Carmona et al., 2002) and the mathematics of DS (Guckenheimer & Holmes, 1983; Alligood et al., 1996), with the tent map or the baker's map famous examples of PL maps introduced to examine chaotic attractors, their fractal geometry, or their topological backbone of infinitely many periodic orbits (Avrutin et al., 2012; 2014; Gardini

& Makrooni, 2019). Switching linear dynamical systems (SLDS) (Ghahramani & Hinton, 2000; Fox et al., 2008; Linderman et al., 2016; 2017; Linderman & Johnson, 2017; Alameda-Pineda et al., 2022) and jump Markov systems (Shi & Li, 2015) capture regime changes by combining multiple linear modes with a discrete switching process. Bayesian extensions (Linderman et al., 2017) allow inference over the number of modes and their transition probabilities, and recent approaches combine SLDS with RNNs (recursive SLDS) to model nonlinear switching behavior (Smith et al., 2021). PLRNNs are another class of systems specifically introduced for DSR (Durstewitz, 2017) and exhibit SOTA performance, with the dendritic PLRNN (Brenner et al., 2022), shallow PLRNN (Hess et al., 2023), or almost-linear RNN (ALRNN; Brenner et al. (2024a)) various representatives of this class. All these models are, however, formulated in *discrete time*, and thus cannot deal with irregularly spaced data or provide solutions at arbitrary time points.

# 3. Model Formulation and Theoretical Analysis

## 3.1. Continuous PLRNN and Solution Method

**Data assumptions** Suppose we are given a data set $\{(t_n, \boldsymbol{x}_n)\}_{n=1}^{T}$ of $T$ observations $\boldsymbol{x}_n \in \mathbb{R}^N$ taken at times $t_n$, potentially sampled at irregular intervals. We assume these data were generated by some latent dynamical process $\boldsymbol{z}(t) \in \mathbb{R}^M$, coupled to the data via an observation (decoder) function $\boldsymbol{x}_n = G(\boldsymbol{z}(t_n))$. $G$ may be linear, an MLP, or in the simplest case just an identity mapping from an $N$-dimensional subspace of the latent space (for the subsequent developments the assumed form for $G$ is not relevant; while more complex en- & decoder models are possible, Brenner et al. (2024b), here we intentionally kept them simple to focus on the computational contributions of the core algorithm).

**Model formulation** Our approach builds on the class of PLRNNs (Durstewitz, 2017; Hess et al., 2023; Brenner et al., 2024a), which use the ReLU as their non-linearity and in discrete time are defined by the map

$$\boldsymbol{z}_{t+1} = \boldsymbol{A}\boldsymbol{z}_t + \boldsymbol{W}\Phi^*(\boldsymbol{z}_t) + \boldsymbol{h}, \qquad (1)$$

where $\boldsymbol{z}_t, \boldsymbol{h} \in \mathbb{R}^M$, $\boldsymbol{A}, \boldsymbol{W} \in \mathbb{R}^{M \times M}$ with $\boldsymbol{A}$ diagonal, and

$$\Phi^*(\boldsymbol{z}_t) = \left( z_t^{(1)}, \ldots, z_t^{(M-P)}, \max\left\{0, z_t^{(M-P+1)}\right\}, \right.$$
$$\left. \ldots, \max\left\{0, z_t^{(M)}\right\} \right)^T. \qquad (2)$$

For $P = M$ (i.e., a ReLU on all latent dimensions) this yields the original vanilla PLRNN (Durstewitz, 2017; Koppe et al., 2019b), while for $P < M$ (i.e., $P$ nonlinear and $M - P$ linear units) we obtain the ALRNN as introduced in Brenner et al. (2024a).

Equation (1) can be rewritten as

$$\boldsymbol{z}_{t+1} = (\boldsymbol{A} + \boldsymbol{W}\boldsymbol{D}_t)\boldsymbol{z}_t + \boldsymbol{h}, \qquad (3)$$

with $\boldsymbol{D}_t := \mathrm{diag}(\boldsymbol{d}_t)$ and $\boldsymbol{d}_t := (1, \ldots, 1, d_t^{(M-P+1)}, \ldots, d_t^{(M)})^T$, such that for $i = M - P + 1 \ldots M$, $d_t^{(i)} = 0$ if $z_t^{(i)} \leq 0$ and $d_t^{(i)} = 1$ otherwise. There are $2^P$ different configurations of the matrix $\boldsymbol{D}_t$, depending on the signs of the coordinates of $\boldsymbol{z}_t$. As a result, the state space is separated into $2^P$ different subregions $\Omega^k$, $k \in \{1, 2, \ldots, 2^P\}$, by $P$ hyperplanes. Within each subregion, the dynamics is governed by the linear map

$$\boldsymbol{z}_{t+1} = \underbrace{(\boldsymbol{A} + \boldsymbol{W}\boldsymbol{D}_{\Omega^k})}_{\boldsymbol{W}_{\Omega^k}} \boldsymbol{z}_t + \boldsymbol{h}, \qquad \boldsymbol{z}_t \in \Omega^k \quad (4)$$

Reformulating the discrete-time PLRNN as a continuous-time system, we obtain an equation as typically used for neural population dynamics (e.g. Song et al. (2016)):

$$\dot{\boldsymbol{z}}(t) = \boldsymbol{A}\boldsymbol{z}(t) + \boldsymbol{W}\Phi^*(\boldsymbol{z}(t)) + \boldsymbol{h} =: \boldsymbol{W}_{\Omega^k}\boldsymbol{z}(t) + \boldsymbol{h}. \quad (5)$$

We call this the *continuous-time PLRNN (cPLRNN)*. In principle, we could train this system just like a Neural ODE, requiring numerical integration for obtaining solutions at particular time points. Instead, here we would like to exploit the system's PL structure and write an analytic solution in each subregion, making numerical integration obsolete and allowing for efficient parallelization of crucial solver steps. This will be the major contribution of the present work.

**Solving the cPLRNN equations** For an invertible and diagonalizable $\boldsymbol{W}_{\Omega^k}$ (with $\mathrm{diag}(\boldsymbol{\lambda}) = \boldsymbol{P}^{-1} \boldsymbol{W}_{\Omega^k} \boldsymbol{P}$), Equation (5) is solved by

$$\boldsymbol{z}(t) = \boldsymbol{P}\,\mathrm{diag}\left(e^{\boldsymbol{\lambda}t}\right) \underbrace{\boldsymbol{P}^{-1}(\boldsymbol{z}_0 + \boldsymbol{W}_{\Omega^k}^{-1}\boldsymbol{h})}_{\boldsymbol{c}} - \boldsymbol{W}_{\Omega^k}^{-1}\boldsymbol{h}$$
$$= \sum_l c^{(l)}\left(e^{\lambda^{(l)}t}\right)\boldsymbol{u}_l - \boldsymbol{W}_{\Omega^k}^{-1}\boldsymbol{h} =: f(t; \boldsymbol{z}_0) \quad (6)$$

with $\boldsymbol{u}_l$ the eigenvector belonging to eigenvalue $\lambda^{(l)}$. The expression for the $i-$th dimension is given by

$$z^{(i)}(t) = \sum_l \underbrace{c^{(l)}u_l^{(i)}}_{\tilde{c}_l} e^{\lambda^{(l)}t} \underbrace{- \left(\boldsymbol{W}_{\Omega^k}^{-1}\boldsymbol{h}\right)^{(i)}}_{\tilde{h}}$$
$$= \sum_l \tilde{c}_l^{(i)} e^{\lambda^{(l)}t} + \tilde{h}^{(i)} =: f^{(i)}(t; \boldsymbol{z}_0) \quad (7)$$

where $\lambda^{(l)} = \mathrm{eigval}(\boldsymbol{W}_{\Omega^k}) \in \mathbb{C}$, $\tilde{h}^{(i)} \in \mathbb{R}$, $\tilde{c}_l^{(i)}$ a constant complex factor, and $\boldsymbol{z}_0 := \boldsymbol{z}(t = t_0) \in \mathbb{R}^M$ is the initial condition. For more details, see Section A.

The solution Equation (7) is only valid as long as the trajectory stays within one linear subregion. To determine the

global solution, one has to determine the "switching times" $t_{\text{switch}}$ for the given initial state $z_0$ and parameters $A, W, h$, i.e. the times at which one of the separating hyperplanes is crossed. In the cPLRNN, this comes down to the first sign change (zero crossing) in one of its ReLU components $z^{(i)}(t), \ i = 1, \ldots, P$:

$$t_{\text{switch}} := \inf\Big\{ t > 0 \,|\, \exists i \in \{1, \ldots, P\} : z^{(i)}(t) = 0$$
$$\wedge \ \text{sgn}\Big(z^{(i)}(t^-)\Big) \neq \text{sgn}\Big(z^{(i)}(t^+)\Big)\Big\} \quad (8)$$

Using Equation (7), one can calculate the roots for the dimensions independently from each other.

**Interval root finding**   Say we are interested in computing solutions $\{\hat{z}_n\}_{n=1}^T$ at time points $\{t_n\}_{n=1}^T$, and from initial state $z_0$ assumed (with no loss of generality) at time $t_0 = 0$. We thus need to search for the first switching time $t_{\text{switch}, 1}$ in the finite interval $(0, t_T)$. Given $t_{\text{switch}, 1}$, we then continue searching within $(t_{\text{switch}, 1}, t_T)$, and so forth; see Section B.1. Since one cannot ensure the search interval $(t_{\text{start}}, t_{\text{end}})$ to be a bracketing interval ($f^{(i)}(t_{\text{start}}) \cdot f^{(i)}(t_{\text{end}}) < 0$; see Figure 4 for an example), and since it is crucial to find the first root rather than just any root, most available root finders are not suited for the problem at hand. We therefore wrote our own root finding algorithm, modeled after the algorithm from the Julia library `IntervalRootFinding.jl` (Sanders et al., 2025). Our version was designed to work well for the specific functional form Equation (7) (sum of exponentials), and for the specific task of finding the *first* root over *several* similar functions. For algorithmic details and the required interval arithmetics, see Section C. Having established a reliable procedure for locating the switching times, the remaining challenge is to perform gradient-descent through these times.

**Differentiating through root times**   We are using stochastic gradient descent to optimize parameters $A, W, h$ on a mean-squared error loss (see Appx. D). Since it is not possible (or extremely tedious) to differentiate through the root finder function because it relies on non-differentiable algorithmic operations, we implemented a customized derivative w.r.t. the parameters $\phi$,

$$\frac{\partial t_{\text{switch}}}{\partial \phi}, \phi \in [A, W, h]. \quad (9)$$

to be used by the automatic differentiation framework.

The switching time is defined implicitly by the constraint

$$f^{(i)}(t_{\text{switch}}(\phi), \phi) := z^{(i)}(t_{\text{switch}}(\phi); \phi) = 0. \quad (10)$$

By the implicit function theorem, if $\frac{\partial f^{(i)}}{\partial t}(t_{\text{switch}}, \phi) \neq 0$, then

$$\frac{\partial t_{\text{switch}}}{\partial \phi} = -\frac{\frac{\partial f^{(i)}}{\partial \phi}(t_{\text{switch}}, \phi)}{\frac{\partial f^{(i)}}{\partial t}(t_{\text{switch}}, \phi)}. \quad (11)$$

Using functional form Equation (7), the time derivative in Equation (11) becomes

$$\frac{\partial f^{(i)}}{\partial t}(t; z_0) = \sum_l \lambda^{(l)} \tilde{c}_l^{(i)} e^{\lambda^{(l)} t} . \quad (12)$$

The parameter derivative $\frac{\partial f^{(i)}}{\partial \phi}$ depends on the gradients of $\tilde{c}_l^{(i)}(\phi)$, $\lambda^{(l)}(\phi)$, and $\tilde{h}^{(i)}(\phi)$, which are computed via automatic differentiation. In the unlikely case of a tangential root, i.e., when the denominator in the implicit derivative vanishes, the gradient is discarded.

**Computing the states**   One great benefit of our method is that in one linear subregion, all states $z(t)$ can be computed analytically and in parallel, in contrast to traditional nonlinear RNNs where this can only be done sequentially, see Algorithm 1 and Section B.2 for an illustration.

---

**Algorithm 1** Computing states $\{\hat{z}_n\}_{n=1}^T$

---

1: **Input:** $\{t_n\}_{n=0}^T$, $z_0 = z(t_0)$, $\phi = \{A, W, h\}$
2: $t_{\text{switch}}, z_s \leftarrow t_0, z_0$         ▷ initial condition
3: $n_b \leftarrow 1$                          ▷ array index
4: $\{\hat{z}\} \leftarrow \{\}$             ▷ initialize trajectory
5: $t_{\text{tot}} \leftarrow t_{\text{switch}}$                ▷ time index
6: **while** $t_{\text{tot}} < t_T$ **do**
7:     $\lambda, \tilde{c}, \tilde{h} \leftarrow \text{PARAMETERS}(\phi, z_s)$    ▷ region param.
8:     $f(\cdot) \leftarrow \text{FUNCTION}(\lambda, \tilde{c}, \tilde{h})$ ▷ define region sol. $f$
9:     $t_{\text{switch}}, z_s \leftarrow \text{ROOT}(f, [t_{\text{tot}}, t_T])$     ▷ switching time
10:    $n_e \leftarrow \max\{i \,|\, t_{\text{switch}} + t_{\text{tot}} > t_i \in \{t_n\}\}$ ▷ region idx
11:    $\{\hat{z}\} \leftarrow \{\hat{z}\} \cup f\big(\{t_n\}_{n=n_b}^{n_e}\big)$     ▷ parallel calculation
12:    $n_b, t_{\text{tot}} \leftarrow n_e + 1, t_{\text{tot}} + t_{\text{switch}}$       ▷ index update
13: **end while**

---

**Sparse teacher forcing (STF)**   In STF, during training model-generated states $z(t)$ are replaced by data-inferred states $\hat{z}(t) = G^{-1}(x(t))$ at time lags $\tau$ chosen based on an estimate of the system's maximal Lyapunov exponent or determined as a hyperparameter (Mikhaeil et al., 2022; Brenner et al., 2022). As shown in Mikhaeil et al. (2022), STF helps in dealing with the exploding and vanishing gradient problem (Bengio et al., 1994) especially when training on chaotic systems, where this problem is an inherent consequence of the system dynamics. Here we employed STF in all directly comparable methods and models (cPLRNN, Neural ODE and standard PLRNN). Note that STF is *only used in model training*, not in testing. For more details, see Section D.

### 3.2. State Space Analysis of Trained Models

While analyzing generic nonlinear DS, for instance identifying their periodic orbits, is generally hard, many of the

required calculations are much more tractable when the system is piecewise-linear. Many of the tools from the smooth linear theory can be adapted to this setting with minor modifications that deal with the behavior of trajectories near the switching manifolds between pairs of PL regions. In this section, we explain how to algorithmically detect equilibria (fixed points) and limit cycles in trained cPLRNNs (see also Section H for a review of further tools as collected in Coombes et al. (2024) that might be useful in the current setting). As before, we assume that the whole state space $\mathbb{R}^M$ is divided into regions $\Omega^k$, $k = 1, \dots, 2^P$, within which the dynamics is linear.

**Equilibria (fixed points)** Equilibria (fixed points) are defined by the condition that the vector field vanishes at these points. Since the system is linear in every subregion, setting Equation (5) to zero, we can solve for $z$ and obtain within each region:

$$z^*_{\Omega^k} = -W^{-1}_{\Omega^k} h. \tag{13}$$

Two scenarios are possible: $z^*_{\Omega^k}$ may either lie *within* region $\Omega^k$ (including its boundary), in which case it is a real fixed point of the system; or it may lie outside of that region, in which case we call it a virtual fixed point. In practice, most fixed points will be virtual. As the number of regions grows exponentially with the number of ReLUs $P$, for large $P$, solving the equation in every region and verifying whether the solution is real or virtual is computationally expensive. In Eisenmann et al. (2023), a heuristic algorithm called SCYFI is presented for discrete PLRNNs that significantly reduces computational costs by using virtual fixed points to initialize the next search run, which under some conditions leads the algorithm to converge in linear time. This algorithm can be easily adapted to the continuous setting by simply changing the fixed-point equation to Equation (13). SCYFI finds fixed points of arbitrary type (stable/ unstable nodes/ spirals, saddles) without the combinatorial explosion that would occur in a naive approach. Figure 1 shows the fixed points (pink) found in a cPLRNN, a standard PLRNN and a ReLU based Neural ODE, trained on the Lorenz-63 system, with the true fixed points overlaid in black.

**Limit cycles** A limit cycle is a closed (but non-constant), isolated orbit of a periodic trajectory of a DS, corresponding to a nonlinear oscillation. In the PL case, it will traverse a sequence of regions, $\mathcal{R} = (\Omega^{k_1}, \dots, \Omega^{k_r})$, before eventually closing up within the initial region, $\Omega^{k_{r+1}} = \Omega^{k_1}$, after one or more iterations. Note that any subregion $\Omega^k$ may occur more than once within this sequence of $r$ subregions, e.g. we may have $\Omega^{k_3} = \Omega^{k_5}$. In order to find the trajectory $\gamma$ of an orbit through $\mathcal{R}$, we may assume that $\gamma$ starts on the switching boundary $\Sigma_{k_r k_1}$. Without loss of generality, assume that $\Sigma_{k_r k_1} = \{z \in \mathbb{R}^M \mid z^{(1)} = 0\}$, so we can write $\gamma(0) \equiv \gamma_0 = (0, y^{(1)}, \dots, y^{(M-1)})^T$. Recall that in each

subregion, the linear ODE has an analytical solution given by Equation (7). Given $\gamma_0$, $\gamma$ will cross $\Sigma_{k_1 k_2}$ at $\gamma(t_1)$ after a time of flight $T_1 \equiv t_1$, reflected in a change of sign in the corresponding coordinate. From there, it continues to evolve through $\Omega^{k_2}$ until it hits $\Sigma_{k_2 k_3}$ at $\gamma(t_2)$ after time of flight $T_2 \equiv t_2 - T_1$. This repeats $r$-times until at the end of region $\Omega^r$, after a total time $T \equiv t_r = \sum_{i=1}^{r} T_i$, the trajectory crosses $\Sigma_{k_r k_1}$ and returns to $\Omega^{k_1}$. At this point, in order to constitute a limit cycle, $\gamma$ needs to satisfy $\gamma(T) = \gamma(0)$, which is by assumption provided for the first dimension and gives $M - 1$ constraints for the remaining dimensions.

In total, the cycle is thus parametrized by $M - 1 + r$ unknowns $y^{(1)}, \dots, y^{(M-1)}, T_1, \dots, T_r$. These are constrained through $M - 1 + r$ equations,

$$
\begin{aligned}
0 &= f^{(d_1)}(T_1; \gamma_0) \\
0 &= f^{(d_2)}(T_2; \gamma(t_1)) \\
&\vdots \\
0 &= f^{(d_r)}(T_r; \gamma(t_{r-1})) \\
y^{(1)} &= f^{(2)}(T_r; \gamma(t_{r-1})) \\
&\vdots \\
y^{(M-1)} &= f^{(M)}(T_r; \gamma(t_{r-1})) ,
\end{aligned} \tag{14}
$$

where $d_i$ is the dimension corresponding to the $i$-th switching event, and $d_r = 1$. Note that Equation (14) holds for *any* type of limit cycle (stable, unstable, or saddle), and by solving it we obtain its *exact* (not approximate) location in state space.

To solve it, we only need an initial guess about the sequence of subregions $\mathcal{R}$ visited and initial estimates for $\{T_i\}$ and $y$, and can then, in principle, use any numerical root finder. While this could also be our root-finding approach from Sect. 3.1, due to its inherently one-dimensional formulation, it would need to be embedded in a multiple-shooting-type scheme. Here we therefore employed a Trust Region solver (Conn et al., 2000) in combination with the Variable Projection method (O'Leary & Rust, 2013) to enhance robustness. Examples of limit cycles obtained this way are depicted in Figure 2, while Figure 6 shows detection on a model trained on simulations with process noise. Fig. 7 further illustrates that limit cycle detection is highly robust with respect to initial misspecification in either $\{T_i\}$ or $y$. Note that detection usually would only be performed *once* after model training and is fast, with runtimes for $M \leq 40$ and $P \leq 30$ all below 15s on 8 cores of an AMD EPYC 9655 CPU.

## 4. Results

In this work we introduce a completely novel type of algorithm for solving PL continuous-time RNNs. The only viable class of direct reference methods we therefore see are Neural ODEs *with ReLU-based activation functions*, tested here with three different solvers (Euler, RK4, Tsit5; see

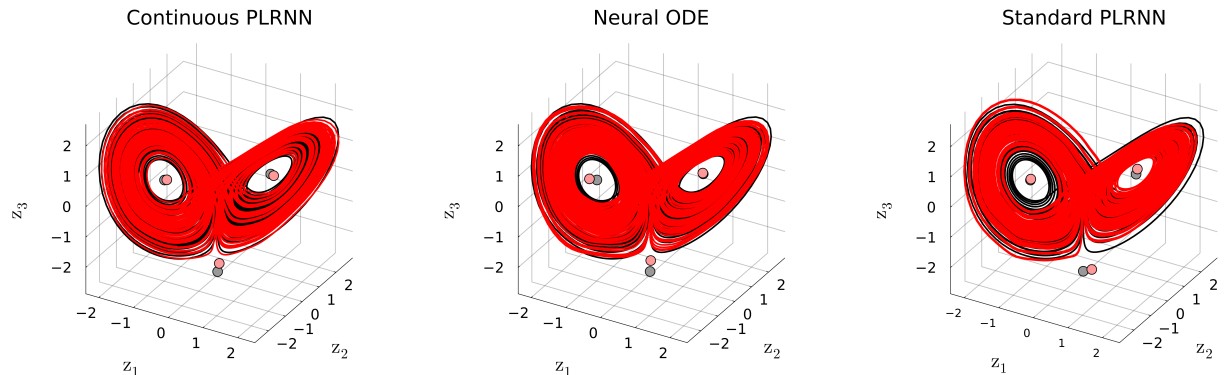

*Figure 1.* Example reconstructions for the three core (ReLU-based) models: Ground truth trajectories and fixed points in black for Lorenz-63 system, and model-generated trajectories ($M = 20$, $P = 10$ for all models) and fixed points found with SCYFI in red.

Appx. E). As a further SOTA reference for DSR (Brenner et al., 2022; 2024a; Hess et al., 2023), we also compared to the *standard, discrete-time PLRNN*, although we emphasize again that its unsuitability for dealing with irregularly spaced and arbitrary time points is exactly one of the weaknesses we wanted to address here. Although less directly relevant for benchmarking our algorithm, we also included two variations of the Neural ODE, namely Latent ODEs (Rubanova et al., 2019) and Neural Controlled Differential Equations (Neural CDEs; Kidger et al. (2020)), in our comparisons, as well as the library-based method SINDy (Brunton et al., 2016). Methods were compared on four benchmarks: the chaotic Lorenz-63 system as a standard benchmark used in the DSR literature, a simulated leaky-integrate-&-fire (LIF) neuron model (Gerstner et al., 2014) as an example system which involves a discontinuity (hard spiking threshold), and membrane potential recordings from a cortical pyramidal neuron and PhysioNet (Goldberger et al., 2000) heartbeat time series as real-world examples (for details, see Appx. F). See Appx. D for hyperparameters.

### 4.1. Performance Measures

For assessing DSR quality, we used two standard measures introduced in the DSR literature for comparing the geometrical and temporal structure of attractors (Koppe et al., 2019b; Mikhaeil et al., 2022): $D_{\text{stsp}}$ is a Kullback-Leibler divergence which quantifies the overlap in attractor geometry by comparing the distributions of true and model-generated trajectories in state space (Brenner et al., 2022) (see Appx. G.1), and $D_{\text{H}}$ denotes the Hellinger distance between true and model-generated power spectra for comparing long-term temporal structure (Mikhaeil et al., 2022; Brenner et al., 2022); see Appx. G.2 for details. We also used the standard mean absolute error (MAE) for short-term prediction performance for the non-chaotic benchmarks (Wood, 2010; Koppe et al., 2019a), and the predictability time (in units of Lyapunov time) for the chaotic benchmark

*Table 1.* Comparison of DSR performance between Neural ODE (with different ODE solvers), cPLRNN, and standard PLRNN (all for $P = 10$; see Tab. 9 for other values of $P$), and for Latent ODE, Neural CDE, and SINDy, on the Lorenz-63 system (all models trained for 2000 epochs). Reported are geometrical ($D_{\text{stsp}}$) and temporal ($D_{\text{H}}$) disagreement in the long-term limit, and valid prediction times, as median $\pm$ median absolute deviation across $10 \times 100$ model trainings $\times$ trajectories, except for [†]Neural ODEs integrated by Euler's method where too many runs diverged (leaving only 4-6 valid model runs). For Latent ODE & Neural CDE only the *best* value across all training runs is provided. Note that SINDy naturally performs best here because it already has the correct function library representing the Lorenz-63 system. Best value in bold, second-best underlined.

| Model | $D_{\text{stsp}} \downarrow$ | $D_{\text{H}} \downarrow$ | Prediction time $\uparrow$ |
|---|---|---|---|
| Neural ODE (Euler) [†] | $0.42 \pm 0.05$ | $0.109 \pm 0.01$ | $0.8 \pm 0.9$ |
| Neural ODE (RK4) | $0.57 \pm 0.29$ | $0.12 \pm 0.04$ | $1.2 \pm 0.8$ |
| Neural ODE (Tsit5) | $0.30 \pm 0.11$ | $0.085 \pm 0.019$ | $\mathbf{1.4 \pm 0.8}$ |
| cPLRNN | $0.37 \pm 0.14$ | $0.116 \pm 0.012$ | $\underline{1.3 \pm 0.8}$ |
| standard PLRNN | $\underline{0.24 \pm 0.06}$ | $\underline{0.079 \pm 0.01}$ | $\mathbf{1.4 \pm 0.9}$ |
| SINDy | $\mathbf{0.17}$ | $\mathbf{0.052}$ | $\underline{1.3 \pm 0.7}$ |
| Neural CDE | $12.05$ | $0.87$ | $0.036 \pm 0.031$ |
| Latent ODE | $14.276$ | $0.84123$ | $0.0 \pm 0.0$ |

(see Appx. G.3).

### 4.2. Chaotic Lorenz-63 System

The celebrated Lorenz (1963) model, originally advanced as a model of atmospheric convection (see Section F.1 for details), was the first and probably most famous example of a system with a chaotic attractor. We trained the cPLRNN, Neural ODE, and the discrete-time PLRNN for number of ReLUs $P = 10$ (see Tab. 9 for further values) and latent dimension $M = 20$, with performance provided in Table 1 and example reconstructions in Figure 1. For all three models, as they were all based on ReLUs, fixed points could also be computed by SCYFI. Note that the reconstruction models were not trained on any data directly indicating the presence of the fixed points, but only on trajectories drawn from the chaotic attractor. Hence, the fixed points

and their position are an inferred feature, constituting a type of topological out-of-domain generalization (Göring et al., 2024). Performance-wise the cPLRNN solutions are on par with the best Neural ODE solver (Tsit5), with none of the differences statistically significant according to Mann-Whitney U tests (all $p > 0.27$), and only slightly worse than those produced by the standard discrete-time PLRNN ($p < 0.043$). SINDy naturally performs best on this particular benchmark, as its polynomial function library matches the Lorenz-63 ground truth equations (as pointed out before, Hess et al. (2023)). In contrast, the bad performance of Latent ODE reflects the fact that it is simply not well suited for generating long-term autonomous roll-outs: Mostly it either diverges, or converges to fixed points, thus failing to reconstruct the long-term behavior of cyclic or chaotic DS. Neural CDE even by design does not reasonably permit auto-regressive roll-outs and hence is per se unsuitable for DSR problems. In contrast to all other models, it was therefore run in data-driven mode during testing (which one may expect to yield an advantage), yet still performs worse than the other methods.

While performance is similar for the the cPLRNN and Neural ODE, the cPLRNN *trains significantly (several times) faster* for comparable performance levels, as evident from Table 2. In fact, the training costs for Neural ODEs have been a major bottleneck so far (Dupont et al., 2019; Ghosh et al., 2020; Fronk & Petzold, 2024). Appx. Fig. 8 further shows that training times for the cPLRNN scale about linearly with latent space dimensionality $M$ and sublinearly with the number of linear subregions. Note that while for large numbers of ReLUs (subregions) $P$ Neural ODEs integrated by forward-Euler start to run faster than the cPLRNN, the forward-Euler method is not a serious alternative as it is well known to diverge on many nonlinear problems (Press et al., 2007), as indicated by its much worse performance on most benchmarks considered here. Moreover, as demonstrated in (Brenner et al., 2024a; Brenner & Koppe, 2026), only a small number of linear subregions may actually be required to reconstruct chaotic attractors or learn various cognitive tasks, indicating that scaling with $P$ is not a practically relevant limitation for the cPLRNN.

*Table 2.* Training time comparison of Neural ODE, cPLRNN, and standard PLRNN for different numbers of PL units $P$ over 1999 epochs (removing the first epoch to eliminate differences due to compile time). Shown are means ± standard deviation [s]. Note that for comparability sequences in each batch were not run in parallel, but sequentially.

| Model | $P = 2$ | $P = 5$ | $P = 10$ |
|---|---|---|---|
| Neural ODE (Euler) | $43.7 \pm 2.3$ | $42.4 \pm 2.0$ | $42.0 \pm 1.9$ |
| Neural ODE (RK4) | $156 \pm 15$ | $158 \pm 12$ | $154 \pm 14$ |
| Neural ODE (Tsit5) | $109 \pm 4$ | $112 \pm 7$ | $107 \pm 5$ |
| cPLRNN | $18.8 \pm 2.9$ | $25 \pm 5$ | $33 \pm 5$ |
| standard PLRNN | $4.65 \pm 0.24$ | $4.68 \pm 0.25$ | $4.72 \pm 0.24$ |

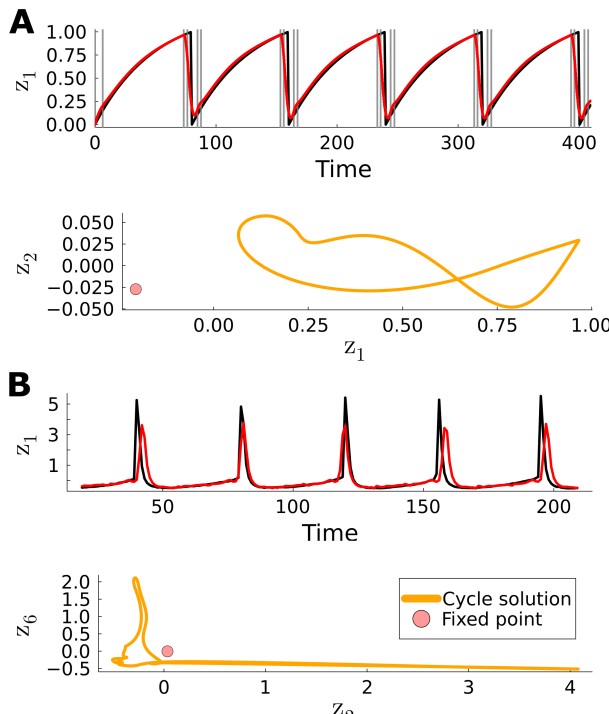

*Figure 2.* A) Top: LIF model (black) and trajectory generated by cPLRNN (red) with $M = 25$ and $P = 2$. The subregion-switching times (gray) in the cPLRNN align well with the spiking times in the LIF model. Bottom: Limit cycle and fixed point found in cPLRNN trained on time series from LIF model. B) Top: Membrane potential recordings (black) and trajectory generated by cPLRNN (red) with $M = 25$ and $P = 6$. Bottom: Limit cycle and fixed point found in cPLRNN trained on empirical data.

### 4.3. Leaky Integrated-and-Fire (LIF) Model

The LIF model is a simple model of a spiking neuron, which describes the temporal evolution of a cell's membrane potential by a linear differential equation, with a hard spiking threshold at which a spike is triggered and the membrane potential is reset (see Appx. F.2 for details; Gerstner et al. (2014)). We used it here as an example for a system with a state discontinuity (note the cPLRNN also has a discontinuity, in its Jacobians). We consider two scenarios, one where we assume we have observations from the system at equally spaced time points ('constant sampling rate'), and one where observations are given at irregular time intervals.

**Equally spaced time points** Figure 2A shows a time series from a cPLRNN ($M = 25$, $P = 2$) trained on trajectories from the spiking LIF model. A limit cycle identified in the trained cPLRNN by solving Equation (14), see Sect. 3.2, and an additional fixed point located by SCYFI, is shown in the state space projection in Fig. 2A, bottom. Fig. 2A further illustrates how the cPLRNN aligns its linear subregion boundaries with the times where most of the 'action happens', i.e. the sudden state resets. Tab. 3 suggests that the

standard PLRNN and the cPLRNN, perform about equally well in this case, as confirmed by a Mann-Whitney U test ($p > 0.05$). Neural ODEs solved by straightforward Euler diverged in 4/10 cases, exposing the limitations of simple explicit solvers in dealing with discontinuities. Likewise, SINDy was not able to reconstruct the LIF behavior.

**Unequally spaced time points** We created a second data set from the LIF simulations with unequally spaced observations by randomly sampling a subset of 10% of the observations. For the standard PLRNN, which cannot naturally deal with irregular temporal intervals, a binning with equal bin sizes $\Delta t = 1$ was created for the irregularly-spaced dataset by linearly interpolating between observations. All performance measures were, however, computed using the full original LIF simulations as comparison template. The results in Table 3 indicate that under these conditions the standard (discrete-time) PLRNN essentially breaks down and clearly loses out performance-wise, highlighting the strengths of a continuous-time approach (see Tab. 7 for another such illustration on a more complex spiking neuron model). Indeed, while the performance of the discrete PLRNN strongly depends on the proportion of available time points, the cPLRNN is (within some limits) hardly affected by it, see Tab. 7. Like the discrete PLRNN, the Latent ODE, Neural CDE, and SINDy essentially were not able to deal with this problem setup, rooted in the issues pointed out in Sect. 4.2.

### 4.4. Electrophysiological Single Neuron Recordings

For a real-world dataset, we chose membrane potential recordings from a cortical neuron (Hertäg et al., 2012). Figure 2B shows time graphs of a cPLRNN ($M = 25, P = 6$) trained on a 6-dimensional embedding (see Appx. F.3) of these time series. A limit cycle corresponding to the spiking activity was identified by solving Equation (14) and, additionally, a stable fixed point: This means the inferred model is *bistable*, with two co-existing attractors, a common observation in cortical neurons (Wang, 2002; Izhikevich, 2007).

We also trained Neural ODEs and a standard PLRNN on the same (embedded) dataset, with performance compared in Table 4. As the table indicates, the cPLRNN significantly outperforms all Neural ODE models on $D_{\text{stsp}}$ and $D_{\text{H}}$ (all $p < 0.022$ according to Mann-Whitney U tests), and is about on par with the standard PLRNN ($p > 0.23$). For Neural ODEs, only results with Tsit5 and RK4 are provided, since for this problem (with fast spiking on top of slower membrane potential variations) Euler always diverged, a well-known issue with Euler's method for stiff ODEs (Press et al., 2007), rendering it unsuitable for a large range of problems. Likewise, Latent ODE, Neural CDE, and SINDy were essentially not able to deal with this real-data example.

### 4.5. Irregularly sampled heartbeat data

As a final real-world example where measurements were taken only at irregular time intervals, we examined time series of heartbeats from the publicly accessible PhysioNet dataset (Goldberger et al., 2000) (see Appx. F.4 for details). The cPLRNN significantly outperformed the discrete PLRNN on forecasting this time series, see Fig. 3, and Tab. 8 for comparisons to the Neural ODE methods on the one time series with the most data points (a full DSR was difficult in this case due to the nature and sparsity of the data). This further supports that beyond DSR, the cPLRNN may also be a strong contender for irregular time series forecasting.

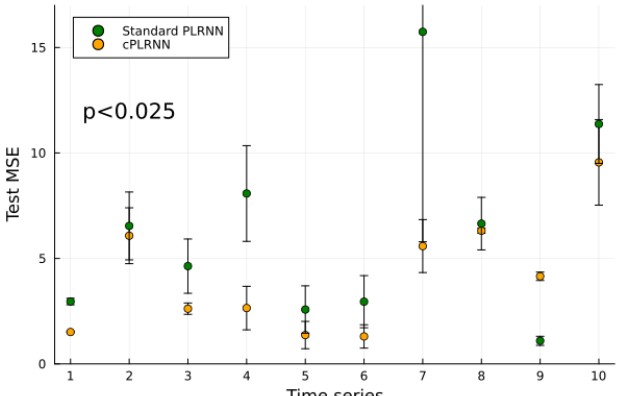

*Figure 3.* Test MSE for models trained on 10 subjects from PhysioNet heartbeat time series, sorted from left-to-right by number of available data points. Shown are medians $\pm$ MAD across all out of 10 training runs that succeeded (for the first time series only 5 training runs were performed). Overall differences in medians across all subjects were statistically significant ($p < 0.025$, Wilcoxon signed ranksum test across the 10 subjects).

## 5. Conclusion

Here we introduce a novel type of algorithm for training and solving PL continuous-time RNNs without the need of numerical integration. Most systems of interest in science and engineering are described in continuous time by sets of differential equations, yet the most successful DSR models are discrete-time maps. Continuous-time reconstruction models are not only a more natural way to describe the temporal evolution in most physical, biological, medical, or engineered systems, but also enable to inter- and extrapolate to arbitrary time points and seamlessly handle observations sampled across irregular temporal intervals. The class of Neural ODEs has been a common choice for dealing with these situations, but Neural ODEs are slow to train (Finlay et al., 2020), lag behind discrete-time models in terms of reconstruction performance (Hess et al., 2023), and are commonly not easily mathematically tractable as we would wish in scientific or medical contexts.

*Table 3.* Performance comparison for Neural ODE, cPLRNN, the standard PLRNN, SINDy, Neural CDE, and Latent ODE trained on regularly and *irregularly* sampled data from the LIF model, assessed after 2000 training epochs. Reported are median $\pm$ MAD for geometrical ($D_{\text{stsp}}$) and temporal ($D_{\text{H}}$) disagreement in limit behavior, and MAE for short-term prediction, across 10 model trainings, [†]except for Neural ODEs integrated by Euler's method where only 6/10 (regular) and 8/10 (irregular) valid runs were produced (all others diverged). For Latent ODE & Neural CDE only the *best* value across all runs is provided. Best value in bold, second-best underlined.

| System | Regularly sampled | | | Irregulary sampled | | |
|---|---|---|---|---|---|---|
| Model | $D_{\text{stsp}} \downarrow$ | $D_{\text{H}} \downarrow$ | MAE $\downarrow$ | $D_{\text{stsp}} \downarrow$ | $D_{\text{H}} \downarrow$ | MAE$\downarrow$ |
| Neural ODE (Euler) [†] | $8.67 \pm 0.12$ | $\mathbf{0.224 \pm 0.049}$ | $\mathbf{0.056 \pm 0.018}$ | $8.88 \pm 0.14$ | $0.272 \pm 0.038$ | $0.082 \pm 0.033$ |
| Neural ODE (RK4) | $8.68 \pm 0.06$ | $0.26 \pm 0.04$ | $\underline{0.075 \pm 0.024}$ | $\underline{8.84 \pm 0.13}$ | $\underline{0.27 \pm 0.04}$ | $\underline{0.074 \pm 0.014}$ |
| Neural ODE (Tsit5) | $8.9 \pm 0.3$ | $0.31 \pm 0.06$ | $0.17 \pm 0.08$ | $8.9 \pm 0.3$ | $0.40 \pm 0.09$ | $0.12 \pm 0.04$ |
| cPLRNN | $\underline{8.65 \pm 0.12}$ | $\underline{0.23 \pm 0.03}$ | $0.076 \pm 0.014$ | $\mathbf{8.71 \pm 0.05}$ | $\mathbf{0.232 \pm 0.024}$ | $\mathbf{0.064 \pm 0.014}$ |
| standard PLRNN | $\mathbf{8.56 \pm 0.07}$ | $\underline{0.23 \pm 0.05}$ | $0.10 \pm 0.04$ | $9.93 \pm 0.18$ | $0.504 \pm 0.087$ | $0.168 \pm 0.047$ |
| SINDy | $11.75$ | $0.94$ | $6.618$ | $10.64$ | $0.839$ | $0.265$ |
| Latent ODE | $10.22$ | $0.66$ | $0.36$ | $11.7$ | $0.85$ | $0.39$ |
| Neural CDE | $11.18$ | $0.85$ | $0.36$ | $11.57$ | $0.86$ | $0.43$ |

*Table 4.* Performance comparison for Neural ODE, cPLRNN, the standard PLRNN, SINDy, Neural CDE, and Latent ODE trained on membrane potential recordings, assessed after 2000 training epochs. Reported are median $\pm$ MAD for geometrical ($D_{\text{stsp}}$) and temporal ($D_{\text{H}}$) disagreement in limit behavior, and MAE for short-term prediction, across 10 model trainings, [†]except diverging runs (2/10 for the standard PLRNN and 1/10 for Neural ODE with RK4; note that this biases results in favor of these models). For Latent ODE & Neural CDE only the *best* value across all training runs is provided. Explicit Euler *always* diverged on this stiff problem. For runs producing equilibira (thus flat power spectra), $D_{\text{H}}$ was set to 1. Best values in bold, second-best underlined.

| Model | $D_{\text{stsp}} \downarrow$ | $D_{\text{H}} \downarrow$ | MAE $\downarrow$ |
|---|---|---|---|
| Neural ODE (RK4) [†] | $2.12 \pm 0.26$ | $0.493 \pm 0.016$ | $\underline{0.65 \pm 0.03}$ |
| Neural ODE (Tsit5) | $2.66 \pm 0.38$ | $0.514 \pm 0.018$ | $\underline{0.65 \pm 0.01}$ |
| cPLRNN | $\underline{1.88 \pm 0.04}$ | $0.474 \pm 0.006$ | $\mathbf{0.627 \pm 0.01}$ |
| standard PLRNN [†] | $\mathbf{1.87 \pm 0.09}$ | $\underline{0.466 \pm 0.017}$ | $0.669 \pm 0.011$ |
| SINDy | $15.28$ | $\mathbf{0.363}$ | $1.618$ |
| Neural CDE | $3.78$ | $0.96$ | $0.99$ |
| Latent ODE | $8.81$ | $0.96$ | $22.24$ |

The cPLRNN addresses these issues by leveraging the PL (ReLU-based) structure to obtain analytic solutions within each linear subregion, avoiding numerical integration and reducing training to the repeated computation of *switching times* at which trajectories cross region boundaries. This enables a semi-analytical forward pass that is both precise and compatible with nonuniform sampling, and runs much faster than integration in Neural ODEs without compromising performance. In addition, the well developed theory for continuous-time PL DS enables to compute important topological properties of trained cPLRNNs, such as their equilibria (fixed points) and limit cycles.

Empirically, cPLRNNs can match the reconstruction quality of discrete-time PLRNNs and Neural ODE baselines on regularly sampled data, while for the irregularly sampled regime, cPLRNNs offer a clear practical advantage by operating directly on the observation times. Beyond ir-

regular sampling and extrapolation to arbitrary time points, cPLRNNs ease state space analysis over their discrete-time counterparts because important geometrical objects, like limit cycles and other invariant sets, are always *continuous* and differentiable, not discrete sets of points, and continuous RNNs enjoy universal approximation guarantees that discrete-time RNNs lack (Sagodi & Park, 2026). cPLRNNs thus provide a step toward continuous-time surrogate models that are both high-performing and amenable to DS analysis, strengthening the role of DSR models as scientific tools rather than purely predictive black boxes.

**Limitations** Training of the cPLRNN is currently not fully numerically robust. In rare cases, optimization terminates due to numerical instabilities, which appear to be related to ill-conditioned eigen-decompositions. Although more stable linear solvers are used in place of explicit matrix inverses, these issues can still occur and may currently somewhat limit reliability for long training runs. As confirmed in Fig. 9, however, even for larger $P$ only a minuscule percentage ($\approx 10^{-5}\%$) of visited subregions is affected by this, making our simple perturbation method (Appx. D.2) a viable solution. Root-finding must be restarted after each switching event, leading to increased computation times in models with many switching boundaries. Caching results of eigen-decompositions may partly mitigate this. Our algorithm for detecting equilibria and limit cycles may miss some of these objects if not properly initialized, but this is a limitation even more severe in all methods relying on numerical continuation (Allgower & Georg, 2003). Finally, we limited our exposition to shallow RNNs – extension to deeper ReLU-based RNNs is possible, but will complicate model analysis and may actually not be necessary for DSR problems (Hess et al., 2023; Brenner et al., 2024a).

**Code** is available at https://github.com/Durst ewitzLab/continuous-time-PLRNN.

## Impact Statement

This paper presents mainly theoretical work to advance a general class of Machine Learning models. Depending on the field of application, there could be many potential societal consequences of our work, none which we feel must be specifically highlighted here.

## Acknowledgements

This work was supported by the German Research Foundation (DFG) through the TRR 265 (subproject A06), individual grant Du 354/15-1 (project no. 502196519), and Du 354/14-1 (project no. 437610067) to DD within the FOR-5159.

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

## A. Functional Expression for State Variables

In order to determine $t_{\text{switch}}$ (and also the state values required inside a linear subregion), we need a functional expression for $z(t)$, i.e. a solution to the linear differential equations 5. In general form, this is given by

$$z(t, z_0) = e^{\boldsymbol{W}_{\Omega^k} t} z_0 + e^{\boldsymbol{W}_{\Omega^k} t} \int_0^t e^{-\boldsymbol{W}_{\Omega^k} \tau} h \, d\tau, \qquad t \in [0, t_{\text{switch}}]. \tag{15}$$

We will assume matrices $\boldsymbol{W}_{\Omega^k}$ to be invertible, as non-invertible matrices constitute a measure-0 subset within the set of matrices and are thus unlikely to occur in training. In this case, the expression for the solution can be simplified to

$$z(t, z_0) = e^{\boldsymbol{W}_{\Omega^k} t}(z_0 + \boldsymbol{W}_{\Omega^k}^{-1} h) - \boldsymbol{W}_{\Omega^k}^{-1} h \,,$$

and in one dimension $i$ and for fixed $z_0$ to

$$z_i(t) = \left(e^{\boldsymbol{W}_{\Omega^k} t}(z_0 + \boldsymbol{W}_{\Omega^k}^{-1} h)\right)^{(i)} \underbrace{-\left(\boldsymbol{W}_{\Omega^k}^{-1} h\right)^{(i)}}_{\tilde{h}} \,,$$

where superscript $(i)$ denotes the $i$-th component of the respective vectors. Furthermore, if $\boldsymbol{W}_{\Omega^k}$ is diagonalizable, $\text{diag}(\boldsymbol{\lambda}) = \boldsymbol{P}^{-1} \boldsymbol{W}_{\Omega^k} \boldsymbol{P}$, this results in

$$z(t, z_0) = \boldsymbol{P} \, \text{diag}\left(e^{\boldsymbol{\lambda} t}\right) \underbrace{\boldsymbol{P}^{-1}(z_0 + \boldsymbol{W}_{\Omega^k}^{-1} h)}_{c} - \boldsymbol{W}_{\Omega^k}^{-1} h = \sum_l c_l \left(e^{\lambda^{(l)} t}\right) u_l - \boldsymbol{W}_{\Omega^k}^{-1} h \,,$$

with $u_l$ the eigenvector corresponding to eigenvalue $\lambda^{(l)}$. The expression for the $i$-th dimension is given by

$$z_i(t) = \sum_l \underbrace{c^{(l)} u_l^{(i)}}_{\tilde{c}_l} e^{\lambda^{(l)} t} \underbrace{-\left(\boldsymbol{W}_{\Omega^k}^{-1} h\right)^{(i)}}_{\tilde{h}} = \sum_l \tilde{c}_l e^{\lambda^{(l)} t} + \tilde{h}^{(i)},$$

i.e. a sum of exponentials with possibly (and most likely) different $\lambda^{(l)}$. As non-diagonalizable matrices will usually occur only rarely, we base our algorithm on the diagonalizability assumption and slightly perturb $\boldsymbol{W}_{\Omega^k}$ to promote distinct eigenvalues in the case of non-diagonalizability, as well as discarding the gradient contributions from these rare cases.

## B. Illustration of Solution Technique

### B.1. Bracketing interval

If there is more than one root present in a given search interval $(t_{\text{start}}, t_{\text{end}})$, it might not be a bracketing interval, i.e. $f(t_{\text{start}}) \cdot f(t_{\text{end}}) < 0$, as illustrated in Section B.1. This means we cannot use any generic root finding algorithm.

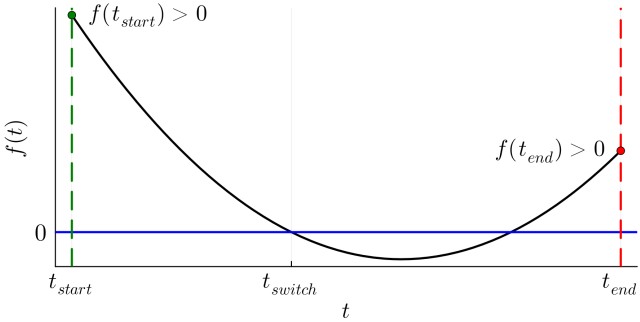

*Figure 4.* Illustration of several roots in a non-bracketing interval.

## B.2. Computing states

For a PLRNN with $P$ ReLU's the state space $\mathbb{R}^M$ is divided into $2^P$ linear subregions, within each of which we have an analytical solution for $z(t)$ according to Equation (6). Assume we would like to obtain a global solution for a trajectory as illustrated in Figure 5, started from an initial condition $z_0$ in subregion $k_1$. As outlined in Algorithm 1, we begin by evaluating Equation (7) with the right $W_{\Omega^k}$ in the first subregion. Based on this, we then compute the first switching time, $t_{s,1}$, and evaluate Equation (7) at all intermediate times provided within the interval $(0, t_{s,1}]$. We then proceed through subsequent subregions in the same manner as indicated below and in Figure 5.

$$
\begin{aligned}
&\{ \overbrace{t_1, t_2, t_3}^{t_{s,1} \leq t < t_{s,1}} , \overbrace{t_4, t_5, t_6}^{t_{s,1} \leq t < t_{s,2}} , \overbrace{t_7, t_8}^{t_{s,2} \leq t < t_{s,3}} \} \\
&\{ \underbrace{z_1, z_2, z_3}_{f_1(t_1, t_2, t_3)}, \underbrace{z_4, z_5, z_6}_{f_2(t_4, t_5, t_6)}, \underbrace{z_7, z_8}_{f_3(t_7, t_8)} \}
\end{aligned}
\tag{16}
$$

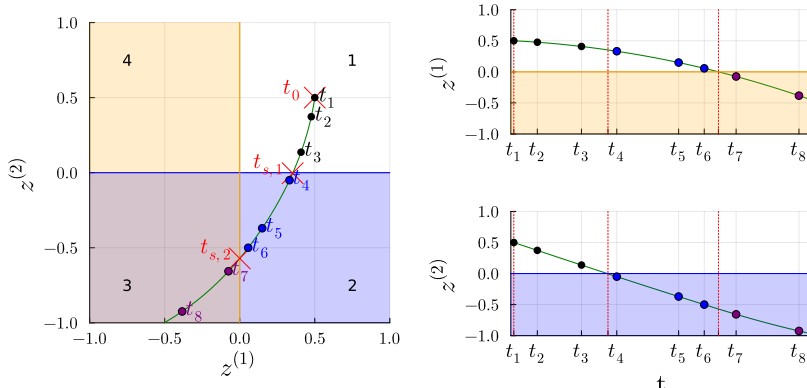

*Figure 5.* Illustration of problem setting: for obtaining a global trajectory solution at arbitrary time points $t_i$ we need to find the switching times $t_{s,k}$ between linear subregions.

# C. The Interval Newton Method

Because our goal is to locate the first root among potentially several roots within a specified interval, the interval Newton method is an appropriate option, as it can identify all roots contained in that interval (Hansen & Sengupta, 1981). Since understanding this method requires familiarity with interval arithmetic, we first provide a brief introduction to that topic.

## C.1. Interval Arithmetic

Interval arithmetic replaces single numbers with intervals and generalizes the usual arithmetic operations so that the outcome is an interval again. For a real-valued function $f$, its interval extension $f_{\text{interval}}(X)$ is defined such that

$$
f(x) \in f_{\text{interval}}(X) \quad \text{for all } x \in X,
\tag{17}
$$

i.e. the image $f_{\text{interval}}(X)$ includes every function value $f(x)$ for each point $x \in X$.

Recall Equation (7), restated here for convenience:

$$
f^{(i)}(t) = \sum_l \tilde{c}_l e^{\lambda^{(l)} t} + \tilde{h}^{(i)}.
\tag{18}
$$

This equation may have complex eigenvalues $\lambda^{(l)}$, but because matrix $W_{\Omega^k}$ itself has real entries, they must appear in complex-conjugate pairs $\lambda^{(l)}, \overline{\lambda^{(l)}}$. Consequently, we can express the corresponding exponentials using real-valued functions as

$$
\tilde{c}_l e^{\lambda^{(l)} t} + \overline{\tilde{c}_l} e^{\overline{\lambda^{(l)}} t} = 2 e^{\operatorname{Re}(\lambda^{(l)}) t} \Big( \operatorname{Re}(\tilde{c}_l) \cos\Big( \operatorname{Im}(\lambda^{(l)}) t \Big) - \operatorname{Im}(\tilde{c}_l) \sin\Big( \operatorname{Im}(\lambda^{(l)}) t \Big) \Big).
\tag{19}
$$

Thus, to construct the interval extension of our function, we require the standard operations—such as addition—together with the interval rules for real exponential, sine, and cosine terms.

**Basic rules of interval arithmetic**  Let $X = [a, b] \subset \mathbb{R}, Y = [c, d] \subset \mathbb{R}$ with $a \leq b$ and $c \leq d$. Then the basic arithmetic rules are given by

- Addition: $X + Y = [a + c,\ b + d]$.

- Subtraction: $X - Y = [a - d,\ b - c]$.

- Scalar Multiplication: For $\mu \in \mathbb{R}$, $\mu X = \begin{cases} [\mu \cdot a,\ \mu \cdot b], & \mu \geq 0, \\ [\mu \cdot b,\ \mu \cdot a], & \mu < 0. \end{cases}$

- Multiplication: $X \cdot Y = [\min\{ac, ad, bc, bd\}, \max\{ac, ad, bc, bd\}]$.

- Division: If $0 \notin Y$, then $1/Y = \left[\frac{1}{d},\ \frac{1}{c}\right]$; if $c < 0 < d$, then $1/Y = \left[-\infty,\ \frac{1}{c}\right] \cup \left[\frac{1}{d},\ \infty\right]$.

**Monotonic functions**  For a monotonic function such as the real-valued exponential function, the interval function evaluation is given by

$$f_{\text{interval}}(X) = \exp([a, b]) = [\exp(a), \exp(b)], \tag{20}$$

i.e. it depends solely on the endpoints of the interval, easing computations.

**Sine and cosine**  To compute the sine and cosine over an interval $X = [a, b]$, one must distinguish and handle several different cases:

1. **$X$ contains a full period:**
   If $b - a \geq 2\pi$ then $X$ contains at least one full period of sine and cosine. In this case, the exact ranges are given by $\sin(X) = [-1, 1], \cos(X) = [-1, 1]$.

2. **$X$ contains both a maximum and a minimum**
   If $X$ contains at least one point of the form $\frac{\pi}{2} + k\pi$ and at least one point of the form $-\frac{\pi}{2} + k\pi$ for some $k \in \mathbb{Z}$, then $\sin(X)$ attains both its global maximum and minimum over $X$, and therefore

   $$\sin(X) = [-1, 1].$$

   An analogous condition holds for $\cos(X)$ if $X$ contains both a point $k\pi$ and a point $\pi + k\pi$.

3. **$X$ contains exactly one extremum**
   If $X$ contains exactly one critical point of the sine function, but not both a maximum and a minimum, then the range is determined by evaluating the function at the endpoints and at the one extremum:

   $$\sin(X) = \left[\min\{\sin(a), \sin(b), \sin(x^*)\}, \max\{\sin(a), \sin(b), \sin(x^*)\}\right],$$

   where $x^* = \frac{\pi}{2} + k\pi$ or $x^* = -\frac{\pi}{2} + k\pi$ is the unique extremum in $X$. An analogous expression holds for $\cos(X)$ with $x^* = k\pi$ or $x^* = \pi + k\pi$.

4. **$X$ contains no extrema**
   If $X$ contains no critical points of the function, then sine or cosine is monotonic on $X$, and the interval evaluation reduces to

   $$\sin(X) = \left[\min\{\sin(a), \sin(b)\}, \max\{\sin(a), \sin(b)\}\right],$$

   and analogously for $\cos(X)$.

### C.2. Interval Newton method

A Newton step in 1D is relatively straightforward. We begin with a chosen point $x_0 \in X$ (in our setting, we take $x_0$ to be the midpoint of $X$; a typical choice). We then compute the "Newton image interval" $N(X)$ as

$$N(X) = x_0 - \frac{f(x_0)}{f'(X)},$$

where $f'$ denotes the first derivative of $f$. Next, we form the intersection of $N(X)$ with the current interval $X$,

$$X_{\text{new}} = N(X) \cap X, \tag{21}$$

and use $X_{\text{new}}$ as the updated interval, since it is guaranteed to contain all roots.

**Existence and uniqueness properties**   The interval Newton method provides strong theoretical guarantees:

- If $X_{\text{new}} = \emptyset$, then the equation $f(x) = 0$ has no solution in $X$.

- If $N(X) \subseteq \text{interior}(X)$, then there exists a unique solution $x^* \in X$.

- If neither condition holds, $X$ may be subdivided and the method applied recursively.

These can be used to define a recursive algorithm to determine the first root in an interval $X$.

**Algorithmic structure** Note that we are interested in the first root over multiple functions $f = \{f^{(i)}\}$ at the same time, and therefore we obtain a list of intervals when performing operations, i.e. $\{N^{(i)}(X)\}$. Our interval Newton algorithm is specified in Algorithm 2 below.

---

**Algorithm 2** Interval Newton step as used in the branch-and-prune root finding described in Section C.3. Initial inputs are $X = [t_{\inf}, t_{\sup}]$, $I_0 = \{M - P + 1 \ldots M\}$

---

**Input:** Interval $X$, candidate dimensions $I_0$
**Output:** Pruning decision, interval $X$, remaining candidate dimensions $I_0$, root candidate $t_{\min}$
**for** $i \in I_0$ **do**
  $\tilde{X}^{(i)} = f^{(i)}(X)$           ▷ compute function image intervals dimension-wise
**end for**
$I_1 := \{ i \in I_0 \mid 0 \in \tilde{X}^{(i)} \}$         ▷ dimensions whose interval images contain zero
**if** $I_1 = \emptyset$ **then**
  **return Prune**, $X, \emptyset, \infty$        ▷ if none of the interval images contain zero $\Rightarrow$ no root
**else**
  **for** $i \in I_1$ **do**
   $\tilde{X}_N^{(i)} \leftarrow N(X^{(i)})$        ▷ compute interval Newton images dimension-wise
   $X_{\text{new}}^{(i)} \leftarrow \tilde{X}_N^{(i)} \cap X$      ▷ intersect Newton images $X_N$ with $X$ to obtain $X_{\text{new}}$
  **end for**
  $I_2 := \{ i \in I_1 \mid X_{\text{new}}^{(i)} \neq \emptyset \}$       ▷ dimensions for which $X_{\text{new}}$ is nonempty
  **if** $I_2 = \emptyset$ **then**
   **return Prune**, $X, \emptyset, \infty$      ▷ for no dimension $X_{\text{new}}$ is nonempty $\Rightarrow$ no root
  **else**
   $I_3 := \{ i \in I_2 \mid X_{\text{new}}^{(i)} \subset \text{interior}(X) \}$   ▷ dims whose intervals $X_{\text{new}}$ are strictly inside $X \Rightarrow$ unique root
   **for** $i \in I_3$ **do**
    $t_{\text{root}}^{(i)} = \text{ROOT}(X_{\text{new}}^{(i)})$      ▷ compute unique root in $X_{\text{new}}^{(i)}$ dimension-wise
   **end for**
   **if** $I_3 \neq \emptyset$ **then**
    $t_{\min} = \min\left\{ \infty, \left\{ t_{\text{root}}^{(i)} \right\}_{i \in I_3} \right\}$     ▷ Determine first root time across all dimensions
   **end if**
   **if** $I_3 = I_2$ **then**
    **return Store**, $X, \emptyset, t_{\min}$     ▷ no remaining candidate dimensions $\Rightarrow$ found minimal root
   **else**
    $I_0 \leftarrow I_2 \setminus I_3$          ▷ remaining candidate dimensions
    $U \leftarrow \bigcup_{i \in I_0} X_{\text{new}}^{(i)}$        ▷ union of intervals $X_{\text{new}}^{(i)}$
    $X \leftarrow [\inf(U), \min\{\sup(U), t_{\min}\}]$      ▷ form new search interval
    **Branch**, $X, I_0, t_{\min}$    ▷ remaining candidate dimensions $\Rightarrow$ bisect $X$ and perform Newton step again
   **end if**
  **end if**
**end if**

---

This branch-and-bound strategy ensures that all solutions in the initial domain are either enclosed or excluded.

## C.3. Branch-and-Prune Root Finding

Our algorithm follows a *branch-and-prune* paradigm with search for rigorous root isolation, just like our model, `IntervalRootFinding.jl`. As we are interested only in the first root, we chose as search order *depth-first*, i.e. the next interval $X$ to be studied is always the one in the tree closest to the left/lower boundary $t_{\inf}$. Given a function $f : \mathbb{R} \to \mathbb{R}^n$ and an initial search region $[t_{\inf}, t_{\sup}] = X \subseteq \mathbb{R}$,

1. **Early Stopping:** As we are only interested in the first root, we can stop the search if $t_{\inf} > t_{\min}$, i.e. if the current root found $t_{\min}$ has a time less than the current interval's lower bound $t_{\inf}$.

2. **Contract:** Perform a Newton step to contract the interval $X$ (see Algorithm 2)

3. **Prune:** The region is empty and the branch is discarded.

4. **Store:** The region contains a unique root candidate and the branch is discarded.

5. **Branch:** If the region remains unknown, it is bisected (by default at a fixed fraction of its width), and the two resulting subregions are returned to the search queue.

### C.4. Numerical issues

The example $f(x) = e^x - 1$ can be used to demonstrate how overflow of floating-point numbers may affect the outcome of root-finding algorithms. Consider a large interval $X = (0, T)$ and the corresponding image $f(X) = (-1, e^T - 1)$. If $e^T$ becomes very large, computing it may cause an overflow, potentially wrapping around to a negative value. This can incorrectly indicate that there is no root in the interval. Imposing a maximum interval length and bisecting intervals that exceed this limit, examining the resulting smaller subintervals first, can mitigate this problem.

Another potential numerical problem is that the switching time $t_{\text{switch}}$ returned by the root finder might not yield exactly $z_{i_{\text{switch}}}(t_{\text{switch}}) = 0$, but values that slightly deviate in either direction, placing the new state slightly before or after the root. This leads to two problems when initializing the system at the new state $z(t_{\text{switch}})$:

- If the state is slightly before the actual boundary crossing, the same root might be discovered again, leading to an infinite loop.

- Even if the value of $z_{i_{\text{switch}}}(t_{\text{switch}})$ is exactly 0, the $D_{\Omega^k}$ matrix is not initialized correctly if the boundary is crossed from negative to positive, since $d_i(t) = 0$ for either $z_i(t) < 0$ or $z_i(t) = 0$. (Autodifferentiation with Zygote does not allow for in-place modifications and therefore we cannot just switch the $d_i$ entry in the $d$-vector to match the new subregion.)

To solve this, we introduce a slight perturbation $\delta t$ to make sure the boundary is actually crossed when we initialize in the next linear subregion.

How to best choose $\delta t$ is still an open question. If chosen too small, the numerical problems may prevail, while if chosen too big, one may potentially jump across further boundaries and thereby alter the dynamics of the system. Here we used a fixed $\delta t = 0.0001$, but one potential future extension is to determine the best value adaptively using ideas from numerical integration (which may be easier in our case since all the derivatives are given in analytical form).

## D. Training Method & Hyper-Parameters

For linking the DSR model (cPLRNN, standard PLRNN, or Neural ODE) with latent states $z_t \in \mathbb{R}^M$ to the actual data $x_t \in \mathbb{R}^N$, we used a simple identity observation model

$$\hat{x}_t = \mathcal{I} z_t \tag{22}$$

with

$$\mathcal{I} = \begin{pmatrix} 1 & 0 & \cdots & 0 & 0 & \cdots & 0 \\ 0 & 1 & \cdots & 0 & 0 & \cdots & 0 \\ \vdots & \vdots & \ddots & \vdots & \vdots & & \vdots \\ 0 & 0 & \cdots & 1 & 0 & \cdots & 0 \end{pmatrix} \tag{23}$$

$$\underbrace{\phantom{xxxxxxxx}}_{N} \underbrace{\phantom{xxxxxxxx}}_{M-N} \tag{24}$$

i.e., the first $N$ dimensions of the latent state $z_t$ were used as read-out neurons.

All models were then trained by sparse teacher forcing (STF; Mikhaeil et al. (2022)) using a standard Mean Squared Error (MSE) loss comparing model predictions $\hat{x}_t$ with actual observations $x_t$,

$$\ell_{\text{MSE}}\left(\{\hat{x}_n\}_{n=1}^T, \{x_n\}_{n=1}^T\right) = \frac{1}{N \cdot T} \sum_{n=1}^T \|\hat{x}_n - x_n\|_2^2 . \tag{25}$$

In our case of an identity observation model, the STF signal is simply given by

$$\tilde{z}_t = \left( \boldsymbol{x}_t, \boldsymbol{z}_t^{(N+1:M)} \right)^T, \tag{26}$$

where the states of the read-out neurons are replaced directly by observations $\boldsymbol{x}_t$ every $\tau$ time steps during training (not at test time, where trajectories evolve freely across the total simulation period!). The STF interval $\tau$ and all other hyperparameters used are reported in Table 5 & Table 6.

*Table 5.* Model hyperparameters for all experiments

| Hyperparameter | Lorenz-63 | LIF regular | LIF irregular continuous | standard | Membrane potential |
|---|---|---|---|---|---|
| Latent dimension $M$ | 20 | 25 | 25 | 25 | 25 |
| PL units $P$ | [2, 5, 10] | 2 | 2 | 2 | 6 |
| Start learning rate | $1.0 \times 10^{-3}$ | $1.0 \times 10^{-3}$ | $1.0 \times 10^{-3}$ | $1.0 \times 10^{-3}$ | $4.0 \times 10^{-3}$ |
| Teacher forcing interval $\tau$ | 16 | 25 | 3 | 25 | 25 |
| Gaussian noise level | 0.05 | 0.05 | 0.05 | 0.05 | 0.05 |
| Sequence length | 200 | 200 | 20 | 200 | 200 |
| State dimension $N$ | 3 | 1 | 1 | 1 | 6 |

*Table 6.* Hyperparameters of training algorithm for all models

| Hyperparameter | Neural ODE | cPLRNN | standard PLRNN |
|---|---|---|---|
| Optimizer | RAdam | RAdam | RAdam |
| Batch size | 16 | 16 | 16 |
| Batches per epoch | 50 | 50 | 50 |
| Epochs | 2000 | 2000 | 2000 |
| End learning rate | $1.0 \times 10^{-5}$ | $1.0 \times 10^{-5}$ | $1.0 \times 10^{-5}$ |
| Gradient clipping norm | 0.0 | 10.0 | 0.0 |
| Solver | [Euler, RK4, Tsit5] | - | - |
| Solver $\Delta t$ | 1.0 | - | - |
| Error tolerance | Default | - | - |
| Observation model $G$ | Identity | Identity | Identity |

## D.1. Hyperparameters for Neural CDE/ Latent ODE

Beyond the crucial core comparisons between the cPLRNN and Neural ODEs and standard PLRNN, for which we made sure they are as comparable as possible in terms of architecture and decoders, we included Latent ODE (Rubanova et al., 2019) and Neural CDE (Kidger et al., 2020), because these models were specifically designed for irregular time series. However, they are much less suited for DSR problems, which were our main focus here, because they do not easily allow for autonomous long-term roll-outs. Especially Neural CDEs, in their original formulation with the default Hermite cubic spline interpolation, create a continuous control path of the observed data by *non-causal* interpolation schemes, i.e. do not allow for autonomous (data-independent) roll-outs at all. Although the rectilinear interpolation scheme proposed in (Morrill et al., 2021) would allow for online predictions, it was not used here since already the original formulation performed poorly, and since it reduces the Neural CDE to a generalized ODE-RNN as described in Rubanova et al. (2019).

The values reported in the tables are the best results over an extensive grid search, varying for Latent ODE (with ODE-RNN encoder) the learning rate $\{1e^{-3}, 1e^{-4}\}$, latent dimension $\{15, 25\}$, recognition ODE state width $\{15, 25\}$, ODE MLP hidden width, GRU network width $\{100, 200\}$, the depths of the recognition and generative ODE MLPs $\{1, 2\}$, batch size $\{1, 8, 16\}$, and observed time point subsampling rate $\{0.35, 0.65\}$, and for Neural CDE the learning rate $\{0.005, 0.001, 0.003, 0.01\}$, batch size $\{8, 16, 32\}$, sequence length $\{15, 25, 50, 80, 100, 200, 400, 800\}$, and hidden-channel/ReLU combinations of $(16, 2)$, $(16, 8)$, $(24, 2)$, $(24, 4)$, $(24, 12)$, $(32, 4)$, $(32, 6)$, $(32, 8)$, $(32, 10)$, $(32, 12)$, $(32, 16)$, $(48, 8)$, $(48, 12)$, $(48, 16)$, $(48, 24)$, $(64, 12)$, $(64, 16)$, $(64, 24)$, and $(64, 32)$.

## D.2. Dealing with non-diagonalizable matrix

To check if we might have encountered a non-diagonalizable matrix $\boldsymbol{W}_{\Omega^k}$, we check if we have duplicate eigenvalues (a necessary condition). We treat two eigenvalues as duplicate if their minimal absolute distance is smaller than a certain threshold ($1e-10$ in our setting).

In case of a non-diagonalizable matrix $\boldsymbol{W}_{\Omega^k}$, we perturb it by adding a matrix $\boldsymbol{W}_{\text{perturb}}$ with entries drawn from a normal distribution ($\mu = 0$, $\sigma = 1$) multiplied with a small prefactor $\delta_{\text{perturb}}$, that in our setting was chosen to be $1e-6$. This resulting matrix is used to compute the weights $\tilde{c}_l$, $\lambda^{(l)}$ and $\tilde{h}^{(i)}$. Furthermore, the respective derivatives of these with respect to the matrix $\boldsymbol{W}_{\Omega^k}$ are discarded (see Algo. 3).

---

**Algorithm 3** PARAMETERS $\boldsymbol{\lambda}, \tilde{\boldsymbol{c}}, \tilde{\boldsymbol{h}}$

---

1: **Input:** $\phi = \{\boldsymbol{A}, \boldsymbol{W}, \boldsymbol{h}\}, \boldsymbol{z}_s$
2: $\boldsymbol{W}_{\Omega^k} \leftarrow \text{REGION\_MATRIX}(\boldsymbol{A}, \boldsymbol{W}, \boldsymbol{z}_s)$
3: **if** DUPLICATE_EIGENVALUES($\boldsymbol{W}_{\Omega^k}$) **then**
4:     $\boldsymbol{W}_{\Omega^k, \text{perturb}} \leftarrow \boldsymbol{W}_{\Omega^k} + \delta_{\text{perturb}} \cdot \boldsymbol{W}_{\text{perturb}}$
5:     $\boldsymbol{\lambda}, \tilde{\boldsymbol{c}}, \tilde{\boldsymbol{h}} \leftarrow \text{PARAMETERS}(\boldsymbol{W}_{\Omega^k, \text{perturb}})$
6:     $\frac{\partial \boldsymbol{\lambda}}{\partial \boldsymbol{W}_{\Omega^k}}, \frac{\partial \tilde{\boldsymbol{c}}}{\partial \boldsymbol{W}_{\Omega^k}}, \frac{\partial \tilde{\boldsymbol{h}}}{\partial \boldsymbol{W}_{\Omega^k}} \leftarrow 0$
7: **else**
8:     $\boldsymbol{\lambda}, \tilde{\boldsymbol{c}}, \tilde{\boldsymbol{h}} \leftarrow \text{PARAMETERS}(\boldsymbol{W}_{\Omega^k})$
9: **end if**

---

## E. Numerical Solvers

We tested three solvers of different numerical complexity for the Neural ODEs, using the Julia (Bezanson et al., 2017) implementation from DifferentialEquations.jl (Rackauckas & Nie, 2017).

**Euler method**    The forward Euler method is a simple first-order scheme that updates the state using a fixed step size $\Delta t$ as

$$z_{n+1} = z_n + f_\theta(t_n, z_n)\Delta t. \tag{27}$$

**Fourth-order Runge-Kutta (RK4)**    The 4th-order Runge–Kutta is a standard explicit method that improves accuracy by combining multiple intermediate evaluations of the vector field:

$$k_1 = f_\theta(t_n, z_n), \tag{28}$$
$$k_2 = f_\theta\left(t_n + \tfrac{\Delta t}{2}, z_n + \tfrac{\Delta t}{2}k_1\right), \tag{29}$$
$$k_3 = f_\theta\left(t_n + \tfrac{\Delta t}{2}, z_n + \tfrac{\Delta t}{2}k_2\right), \tag{30}$$
$$k_4 = f_\theta(t_n + \Delta t, z_n + \Delta t k_3), \tag{31}$$

followed by the update

$$z_{n+1} = z_n + \tfrac{\Delta t}{6}(k_1 + 2k_2 + 2k_3 + k_4). \tag{32}$$

The Julia implementation uses a defect control as described in Shampine (2005).

**Tsitouras 5/4 method (Tsit5)**    Tsit5 is an explicit adaptive Runge–Kutta method of order five. It automatically adjusts the step size to control a local error by comparing 5th and 4th-order solution, yielding an efficient trade-off between accuracy and computational cost (Tsitouras, 2011).

## F. Benchmark Systems

### F.1. Lorenz63

The *Lorenz-63 system* (Lorenz, 1963) is a continuous-time dynamical model that was initially introduced as a minimal model of atmospheric convection. It provides the time evolution of three state variables through a set of nonlinear differential equations,

$$\frac{dx_1}{dt} = \sigma(x_2 - x_1),$$
$$\frac{dx_2}{dt} = x_1(\rho - x_3) - x_2,$$
$$\frac{dx_3}{dt} = x_1 x_2 - \beta x_3,$$

where $x_1$, $x_2$, and $x_3$ represent, respectively, the convection rate, the horizontal temperature difference, and the vertical temperature difference. The parameters $\sigma$, $\rho$, and $\beta$ are physical constants associated with the Prandtl number, the Rayleigh number, and the geometric configuration of the system. For particular parameter choices, such as $\sigma = 10$, $\rho = 28$, and $\beta = \frac{8}{3}$, the system exhibits chaotic behavior. For these parameters specifically, the famous "butterfly attractor" emerges, a canonical illustration of deterministic chaos in low-dimensional DS. We simulated a trajectory of $T = 10^5$ time steps, using the DynamicalSystems.jl Julia library (Datseris, 2018). The time step for the standard ALRNN was set to $\mathrm{d}t = 10^{-2}$. To obtain a comparable configuration for the continuous models, we scaled the time values $\{t_n\}$ by a factor of 100.

### F.2. Leaky Integrate-and-Fire (LIF) Neuron

The LIF neuron (Gerstner et al., 2014) is a simple continuous-time model that mimics the membrane potential dynamics of a spiking neuron, with spikes 'pasted' on top whenever a spiking threshold is crossed. The membrane potential $V(t) \in \mathbb{R}$ evolves according to the linear differential equation

$$\tau \frac{dV(t)}{dt} = -V(t) + RI(t),$$

where $I(t)$ is some input current, $R$ the membrane resistance, $C$ the membrane capacitance, and $\tau = RC$ the membrane time constant. Each time the membrane potential crosses a fixed threshold $V_{\text{th}}$, a spike is triggered and the membrane potential is reset to $V_{\text{reset}}$. Since this is a simple $1d$ linear ODE, for given $I(t)$ the model can just be integrated analytically. We simulate trajectories with $T = 10^3$ time steps, using parameters $R = 5, C = 10^{-3}, V_{\text{th}} = 1$, and reset value $V_{\text{reset}} = 0$. A constant input current $I(t) = 0.25$ was used, producing regular spiking.

### F.3. Electrophysical Single neuron Recordings

As an empirical dataset, we employ electrophysiological recordings obtained from a cortical neuron (Hertäg et al., 2012). For empirical data, for which the dimensionality of the underlying DS is often (much) higher than the dimensionality of the observation space (as is this case here where only scalar voltage recordings were available), it is necessary to extend the observation space for the purpose of reconstruction to a higher-dimensional embedding space where trajectories and their derivative directions are sufficiently resolved and smooth (Kantz & Schreiber, 2004). The most common technique for this is the method of temporal delay embedding (Takens, 1981; Sauer et al., 1991). For a one-dimensional observed time series $\{x_t\}$ as here, a $d$-dimensional embedding is defined by

$$\mathbf{x}_t^{\text{emb}} = \left(x_t, x_{t-\tau}, \ldots, x_{t-(d-1)\times\tau}\right)^\top, \tag{33}$$

where $\tau$ is a time lag, typically inferred from the autocorrelation function of the corresponding time series. Here we used $d = 6$ and a time lag of $\tau = 13$.

### F.4. Heartbeat dataset

We used heart rate (HR) data from the PhysioNet Computing in Cardiology Challenge 2012 dataset (`https://phys ionet.org/content/challenge-2012/1.0.0/`) (Goldberger et al., 2000). We selected the ten subjects from set A with the densest HR sampling. Non-negative measurements were retained and timestamps were converted to hours, yielding an irregularly sampled scalar time series. Each series was delay-embedded into three dimensions as described above. Delay coordinates were obtained by linear interpolation on the original time grid. The lag $\tau$ was set to four times the mean inter-measurement interval and clipped to the interval [0.5,2.0] hours. The embedded trajectory was split into training (80%) and test (20%) segments. Each embedding coordinate was z-scored using the mean and standard deviation computed on the training segment only. The corresponding irregular observation times were stored alongside the normalized trajectories for continuous-time model training.

## G. Evaluation Metrics

### G.1. Geometrical Measure: $D_{\text{stsp}}$

Given probability distributions $p(\boldsymbol{x})$ over ground-truth trajectories and $q(\boldsymbol{x})$ over model-generated trajectories in the state space, $D_{\text{stsp}}$ is defined as the Kullback-Leibler (KL) divergence

$$D_{\text{stsp}} := D_{\text{KL}}(p(\boldsymbol{x}) \parallel q(\boldsymbol{x})) = \int_{\mathbf{x} \in \mathbb{R}^N} p(\boldsymbol{x}) \log \frac{p(\boldsymbol{x})}{q(\boldsymbol{x})} \, \mathrm{d}\boldsymbol{x}. \tag{58}$$

For low-dimensional observation spaces, $p(\boldsymbol{x})$ and $q(\boldsymbol{x})$ can be estimated via histogram-based binning procedures (Koppe et al., 2019a; Brenner et al., 2022), in which the Kullback–Leibler divergence is approximated by

$$D_{\text{stsp}} = D_{\text{KL}}(\hat{p}(\boldsymbol{x}) \parallel \hat{q}(\boldsymbol{x})) \approx \sum_{k=1}^{K} \hat{p}_k(\boldsymbol{x}) \log \frac{\hat{p}_k(\boldsymbol{x})}{\hat{q}_k(\boldsymbol{x})}. \tag{59}$$

Here, $K = m^N$ denotes the total number of bins, with $m$ bins allocated to each dimension. The quantities $\hat{p}_k(\boldsymbol{x})$ and $\hat{q}_k(\boldsymbol{x})$ represent the normalized counts in bin $k$ corresponding to the ground-truth and model-generated orbits, respectively.

To compute $D_{\text{stsp}}$ for the 1d time series in Tab. 3 and Tab. 4, 3d delay-embeddings were used to sufficiently unfold the cycles in state space with with lag $\tau = 17$ for the LIF neuron and $\tau = 13$ for the real membrane potential recordings, repetively. Moreover, since the Kullback-Leibler divergence $D_{\text{stsp}}$ is a distributional measure but limit cycles, unlike chaotic attractors, do not naturally come with a distribution, Gaussian noise ($\mathcal{N}(0, 0.1)$) was added to both ground truth and generated trajectories to make this measure applicable.

### G.2. Temporal Measure: $D_{\mathrm{H}}$

To quantify the long-term temporal agreement we compare the power spectra between true and reconstructed DS, employing the *Hellinger distance* (Mikhaeil et al., 2022; Hess et al., 2023). Let $f_i(\omega)$ and $g_i(\omega)$ be normalized power spectra of the $i$-th variable of the observed ($\boldsymbol{X}$) and generated ($\boldsymbol{X}_R$) time series, respectively, where $\int_{-\infty}^{\infty} f_i(\omega)d\omega = 1$ and $\int_{-\infty}^{\infty} g_i(\omega)d\omega = 1$, then the Hellinger distance is given by

$$H(f_i(\omega), g_i(\omega)) = \sqrt{1 - \int_{-\infty}^{\infty} \sqrt{f_i(\omega)g_i(\omega)}\, d\omega} \tag{34}$$

with $H(f_i(\omega), g_i(\omega)) \in [0, 1]$, where 0 indicates perfect alignment.

In practice, the Hellinger distance is computed using fast fourier transforms (FFT; Cooley & Tukey (1965)), which results in $\hat{\boldsymbol{f}}_i = |\mathcal{F}x_{i,1:T}|^2$ and $\hat{\boldsymbol{g}}_i = |\mathcal{F}\hat{x}_{i,1:T}|^2$, where the vectors $\hat{\boldsymbol{f}}_i$ and $\hat{\boldsymbol{g}}_i$ represent the discrete power spectra of the ground truth traces $x_{i,1:T}$ and the model-generated traces $\hat{x}_{i,1:T}$, respectively. Because raw power spectra are typically quite noisy, they are smoothed by applying a Gaussian filter with standard deviation $\sigma_s$. The resulting spectra are normalized to fulfill $\sum_\omega \hat{f}_{i,\omega} = 1$ and $\sum_\omega \hat{g}_{i,\omega} = 1$. $H$ is then computed as

$$H(\hat{\boldsymbol{f}}_i, \hat{\boldsymbol{g}}_i) = \frac{1}{\sqrt{2}} \left\| \sqrt{\hat{\boldsymbol{f}}_i} - \sqrt{\hat{\boldsymbol{g}}_i} \right\|_2 , \tag{35}$$

with element-wise square root. Finally we average $H$ across dimensions to obtain $D_{\mathrm{H}}$:

$$D_{\mathrm{H}} = \frac{1}{N} \sum_{i=1}^{N} H(\hat{\boldsymbol{f}}_i, \hat{\boldsymbol{g}}_i) \tag{36}$$

with hyperparameter $\sigma_s$.

For the comparisons on the Lorenz-63 as well as the cortical neuron dataset, we used $\sigma_s = 20$, while for the LIF model comparisons no smoothing was necessary.

### G.3. Valid Prediction Times

For determining the prediction times, the MSE between the generated ($\{\hat{z}_t\}$) and the ground truth trajectories ($\{z_t\}$) was computed and the minimal time $T_{\text{predict}}$ assessed, for which the deviation was larger than a threshold $\epsilon_{\text{predict}}$, here chosen to be 0.05:

$$T_{\text{predict}} = \min\left(t \mid \|\hat{z}_t - z_t\|^2 > \epsilon_{\text{predict}}\right) . \tag{37}$$

This was normalized by the max. Lyapunov exponent $\lambda_{\max} \approx 0.9056$, scaled to units of step size $\Delta t = 0.01$.

For the values in Table 1 and Table 9, we present the mean and standard deviation over 100 time series, with the ground truth time series the same for all models.

## H. Mathematical Tools for Analyzing Trained Piecewise-Linear Systems

### H.1. Classification of PL Systems

While always linear and hence smooth within regions, PL systems can exhibit different degrees of non-smoothness locally. Following previous literature (Leine & Nijmeijer, 2013; Bernardo et al., 2008), (Coombes et al., 2024) classify systems by their degree of discontinuity across switching manifolds as follows:

1. *Continuous PL systems*, such as the class of cPLRNNs considered here, have continuous states and vector fields, but different Jacobians on both sides, i.e., the vector field is not smooth.

2. *Filippov systems* have continuous states, but discontinuous vector fields. On the boundary, the vector field can be (non-uniquely) interpolated as a convex combination of the vector fields on both sides. An example of this type of system is the McKean model.

3. *Impact systems* are discontinuous in both states and vector fields. The behavior of a trajectory that hits the boundary at time $t_0$ can be described by a jump operator $\mathcal{J}$, $\lim_{t \downarrow t_0} x(t) = \mathcal{J}(\lim_{t \uparrow t_0} x(t))$. An example in two dimensions is the planar leaky integrate-and-fire model.

## H.2. Stability Analysis of Periodic Orbits

In smooth systems, Floquet theory can be used to study the stability and bifurcations of periodic orbits. The starting point for this theory is the equation

$$\dot{\boldsymbol{\Phi}} = \mathrm{D}f(\boldsymbol{x}^\gamma(t))\boldsymbol{\Phi}, \quad \boldsymbol{\Phi}(0) = \boldsymbol{I}_m, \tag{38}$$

where $\dot{\boldsymbol{x}} \equiv f(\boldsymbol{x})$, $\boldsymbol{x}^\gamma$ is a periodic orbit of period $T$, $\boldsymbol{\Phi}$ is the fundamental matrix of the ODE, whose columns are linearly independent solutions of the system, and $I_m$ is the identity matrix in $m$ dimensions. The eigenvalues $\lambda_k$ of the monodromy matrix $\boldsymbol{\Phi}(T)$ are called *Floquet multipliers*, and writing them as $\lambda_k = e^{\kappa_k T}$ defines *Floquet exponents* $\kappa_k$. These exponents play a similar role in the study of periodic orbits as Lyapunov exponents do in the study of chaotic systems and are foundational for the derivations in Coombes et al. (2024).

In order to make use of Floquet theory, since we are in a non-smooth setting, we need to make some adjustments and introduce a Floquet theory of PL systems. The key ingredient for this purpose are so-called *saltation operators*; these quantify how a perturbation of a trajectory – in our case, a periodic orbit $\gamma : [0, T] \to \mathbb{R}^m$ – behaves when crossing a switching manifold. Here, a perturbation can be viewed as a vector field along the curve, $\delta \boldsymbol{x} : [0, T] \to T\mathbb{R}^m \cong \mathbb{R}^m$, which evolves according to

$$\frac{\mathrm{d}}{\mathrm{d}t}\delta \boldsymbol{x} = \boldsymbol{W}_{\Omega^k}\delta \boldsymbol{x} \tag{39}$$

in region $\Omega^k$. Hence, the saltation operator $S_k$ can be represented as an $m \times m$ matrix that acts linearly on the tangent space of perturbations,

$$\delta \boldsymbol{x}(t_i^+) = \boldsymbol{S}_k(t_i)\delta \boldsymbol{x}(t_i^-). \tag{40}$$

In Coombes et al. (2024), an explicit expression for the saltation operator of a boundary crossing is derived in terms of the Jacobian of the jump operator (if defined), the velocity of the curve, and the gradient of the indicator function defining the switching manifold. With this tool at hand, the monodromy matrix of the smooth theory can be replaced with a new matrix that is defined by following the orbit, for each segment multiplying the corresponding propagation matrix $G$, and for each boundary-crossing multiplying the saltation operator from the left. The Floquet multipliers are then obtained as the non-trivial eigenvalues of that matrix, and the orbit is linearly stable if all multipliers are smaller than 1 in absolute value.

## H.3. Advanced Tools

Coombes et al. (2024) further consider systems of coupled oscillators and introduce tools to analyze the stability of periodic orbits in synchronized systems of such oscillators. The formalisms introduced are based on the basic tools described above and include *infinitesimal phase* and *isostable response* (iPRC and iIRC[1]) and the *master stability function*. Each of these tools has been well-established in the literature for smooth systems and is extended to the non-smooth setting by the use of saltation operators.

To give an idea of these tools, here we provide a quick overview of the phase and amplitude responses for autonomous, non-coupled systems: $\dot{\boldsymbol{x}} = f(\boldsymbol{x})$, $\boldsymbol{x} \in \mathbb{R}^m$. A neighborhood of a (hyperbolic) limit cycle in such a system can be reparameterized in terms of a phase coordinate $\theta$ specifying progression along the cycle and amplitude coordinates $\psi_j$, $j \in \{1, \ldots, m-1\}$ quantifying orthogonal deviation from the cycle; these coordinates evolve as

$$\dot{\theta} = \omega, \quad \dot{\psi}_j = \kappa_j \psi_j, \tag{41}$$

where $\omega = \frac{2\pi}{T}$ is a constant angular velocity and $\kappa_k$ are the Floquet exponents from the previous Sect. Curves of constant phase are called *isochrons*, those of (individual) constant amplitude coordinates *isostables*. We denote the functions that reparameterize the neighborhood as $\Theta(\boldsymbol{x}) = \theta$ and $\Sigma_k(\boldsymbol{x}) = \psi_k$. From these functions, one defines the *infinitesimal phase response*,

$$\boldsymbol{\mathcal{Z}} := \nabla_{\boldsymbol{x}^\gamma}\Theta, \tag{42}$$

---

[1]The "C" historically stands for "curve".

which quantifies how the phase along the cycle reacts to small perturbations and can be obtained as the solution of the initial value problem,

$$\dot{\boldsymbol{\mathcal{Z}}} = -\mathrm{D}f(\boldsymbol{x}^{\gamma}(t))^{\top}\boldsymbol{\mathcal{Z}}, \quad \boldsymbol{\mathcal{Z}}(0) \cdot f(\boldsymbol{x}^{\gamma}(0)) = \omega. \tag{43}$$

Similarly, the *infinitesimal isostable response*,

$$\dot{\boldsymbol{\mathcal{I}}}_k := \nabla_{\boldsymbol{x}^{\gamma}}\Sigma_k, \tag{44}$$

describes the response of amplitude coordinates under small perturbations and evolves as

$$\dot{\boldsymbol{\mathcal{I}}}_k = \left(\kappa_k\boldsymbol{\mathcal{I}}_k - \mathrm{D}f(\boldsymbol{x}^{\gamma}(t))^{\top}\right)\boldsymbol{\mathcal{I}}_k, \quad \boldsymbol{\mathcal{I}}_k(0) \cdot \boldsymbol{v}_k = 1, \tag{45}$$

$\boldsymbol{v}_k$ being the eigenvector corresponding to $\kappa_k$.

In Coombes et al. (2024), it is shown that, across switching manifolds, these functions behave as

$$\lim_{t \downarrow t_i} \boldsymbol{\mathcal{Z}}(t) = (\boldsymbol{S}^{\top}(t_i))^{-1} \lim_{t \uparrow t_i} \boldsymbol{\mathcal{Z}}(t) \quad \text{and} \quad \lim_{t \downarrow t_i} \boldsymbol{\mathcal{I}}_k(t) = (\boldsymbol{S}^{\top}(t_i))^{-1} \lim_{t \uparrow t_i} \boldsymbol{\mathcal{I}}_k(t), \tag{46}$$

where, as before, $\boldsymbol{S}$ is the saltation operator at that boundary. From this and the forms of Equation (43) and Equation (45), one can easily see that iPRCs and iIRCs admit closed-form solutions consisting of products of matrix exponentials and inverse-transposed saltation operators Equation (46), whose initial conditions can be determined by requiring periodicity and the respective normalization conditions.

## I. Additional Figures

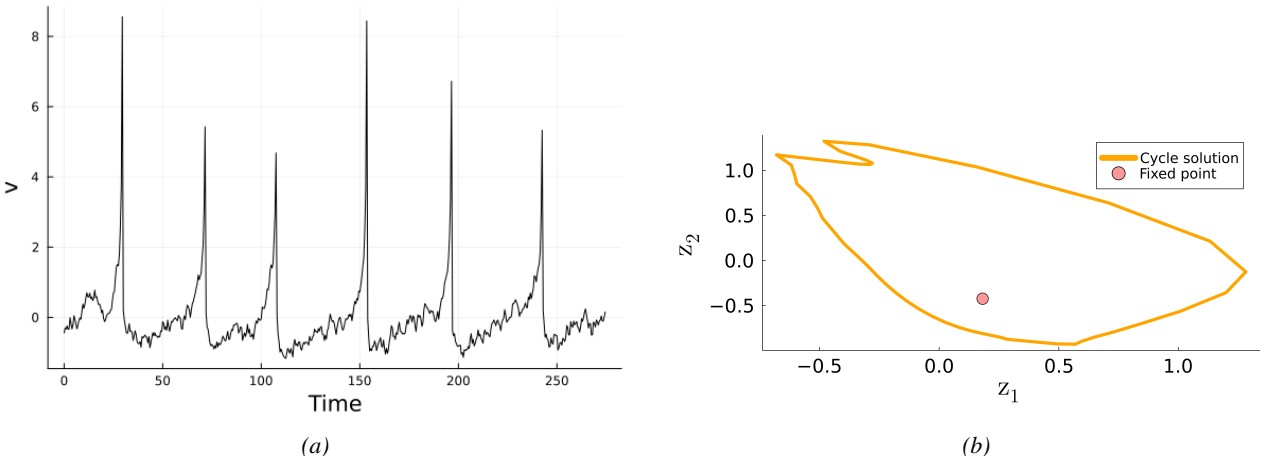

*(a)*                  *(b)*

*Figure 6.* (a) Time series from the (Izhikevich, 2003) spiking neuron model (parameters $a = 0.02$, $b = 0.2$, $c = -65.0$, $d = 8.0$, $I = 10.0$), simulated as SDE with process noise of $\sigma = 2.0$. (b) Limit cycle and fixed point identified semi-analytically, using the methods in sect. 3.2, in a cPLRNN ($P = 2$, $M = 30$) trained on data irregularly sampled from this model.

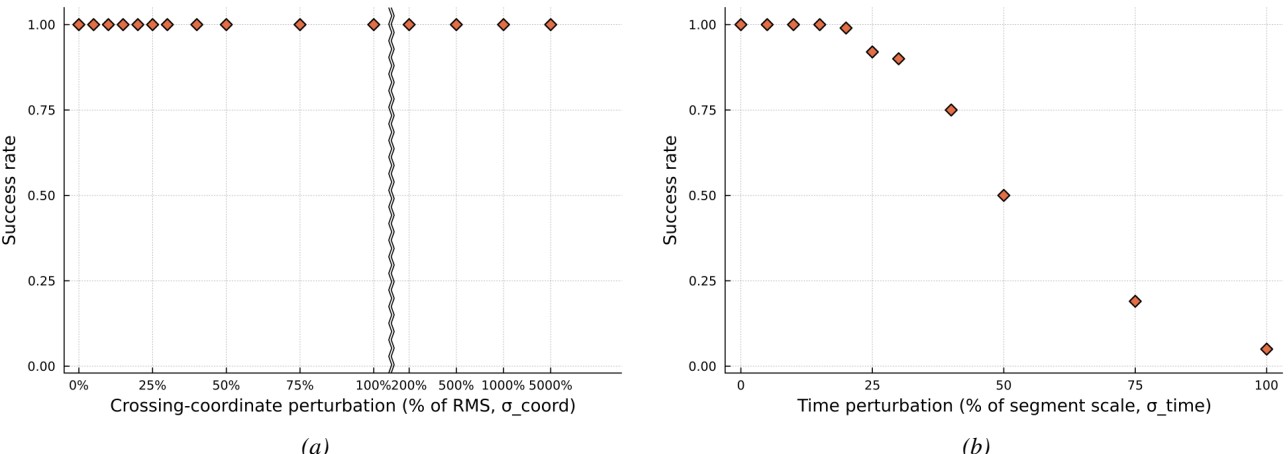

*Figure 7.* Scaling and robustness analysis for limit cycle detection. Experiments were conducted on a 22-region cycle based on the Lorenz system. Each data point represents 100 independent runs of the algorithm. Left: Crossing coordinates were perturbed additively as $x'_i = x_i + \sigma_x s_x \xi_i$, where $\xi_i \sim \mathcal{N}(0, I_{n-1})$ and $s_x$ is the global RMS magnitude of all reduced crossing coordinates. Right: Segment times of flight were perturbed multiplicatively as $\tilde{\tau}_i = \tau_i \exp(\sigma_t \eta_i)$, with $\eta_i \sim \mathcal{N}(0, 1)$, yielding lognormal relative perturbations, $\log(\tilde{\tau}_i / \tau_i) \sim \mathcal{N}(0, \sigma_t^2)$.

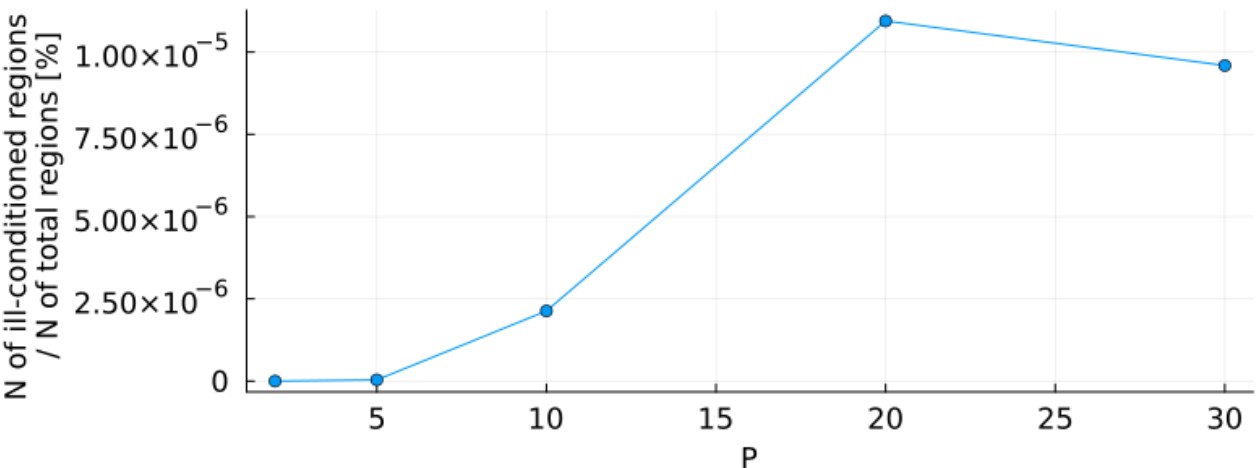

*Figure 8.* Analysis of scaling behavior. Top: Mean runtime per epoch for one training run per setting. Left: Variation of $M$ for fixed $P = 2$; right: variation of $P$ for fixed $M = 40$. The visually apparent absence of any scaling with $P$ for the Neural ODE integrated by forward-Euler was statistically confirmed by a lack of significant correlation ($p = 0.2499$). Bottom: Runtime per epoch vs. total number of regions visited in the epoch for one training run with fixed $M = 40$ and varying $P$ values.

*Figure 9.* Ratio of ill-conditioning matrices $W_\Omega$ among the total number of encountered subregions (duplicates are counted separately) during one training run for different $P$ values for fixed $M = 40$. For fixed $P = 2$ and varying $M \in \{5, 10, 20, 30, 40\}$, the ratio was always zero.

## J. Additional Tables

*Table 7.* Performance of cPLRNN vs. standard PLRNN for different percentages of available sampling points for the LIF example under *irregular* sampling, and performance comparison on another synthetic example, the Izhikevich neuron with *irregular* sampling from the cyclic regime. Values for $D_{\text{stsp}}$ and $D_H$ are given as median $\pm$ MAD. The apparent decrease in cPLRNN performance when increasing data size from 10% to 20% was statistically non-significant (Mann-Whitney U-test; $D_{\text{stsp}}$: $U = 15$, 2-sided $p = 0.2544$; $D_H$: $U = 20$, 2-sided $p = 0.5941$). Note that the $D_{\text{stsp}}$ was calculated in 1d here.

| Model | $D_{\text{stsp}} \downarrow$ | | | $D_H \downarrow$ | | |
| | 10% LIF | 20% LIF | Izhikevich | 10% LIF | 20% LIF | Izhikevich |
|---|---|---|---|---|---|---|
| cPLRNN | $\mathbf{0.26 \pm 0.03}$ | $\mathbf{0.33 \pm 0.25}$ | $\mathbf{7.8 \pm 0.3}$ | $\mathbf{0.232 \pm 0.024}$ | $0.26 \pm 0.5$ | $\mathbf{0.746 \pm 0.003}$ |
| standard PLRNN | $4.3 \pm 0.3$ | $0.7 \pm 0.3$ | $8.6 \pm 2.7$ | $0.5 \pm 0.3$ | $\mathbf{0.15 \pm 0.9}$ | $1$ |

*Table 8.* Performance comparison on *irregularly sampled heartbeat time series* from the PhysioNet/Computing in Cardiology Challenge 2012 (https://physionet.org/content/challenge-2012/1.0.0/). The subject with the most data points was chosen for this comparison, and the time series delay-embedded ($\tau = 1$ hour, $d = 3$), normalized and split into a training (80%) and test trajectory (20%), with MSE evaluated on the test set. Values for cPLRNN vs. standard PLRNN are across 5 runs (where for the standard PLRNN linear interpolation to full hours was used, see sect. 4). Neural ODE trained by forward-Euler quickly diverged on this problem, leading to the large values.

| Measure | Neural ODE (Euler) | Neural ODE (RK) | Neural ODE (Tsit5) | cPLRNN | standard PLRNN |
|---|---|---|---|---|---|
| MSE (test) $\downarrow$ | $4e^3 \pm 6e^3$ | $1.57 \pm 0.15$ | $1.8 \pm 0.3$ | $\mathbf{1.512 \pm 0.018}$ | $2.96 \pm 0.15$ |

*Table 9.* Comparison of reconstruction performance between Neural ODE (with different ODE solvers), continuous PLRNN (cPLRNN), and standard PLRNN for different latent dimensions $P$ for models trained for 2000 epochs on the Lorenz-63 dataset. Reported are geometrical ($D_{\text{stsp}}$) and temporal ($D_H$) disagreement in the limit (lower is better) as median $\pm$ median absolute deviation across 10 model trainings, except for [†]Neural ODEs integrated by Euler's method where too many runs diverged (leaving only 4-6 valid model runs).

| Model | $P = 2$ | | $P = 5$ | | $P = 10$ | |
| | $D_{\text{stsp}}$ | $D_H$ | $D_{\text{stsp}}$ | $D_H$ | $D_{\text{stsp}}$ | $D_H$ |
|---|---|---|---|---|---|---|
| Neural ODE (Euler) [†] | $14.7 \pm 0.7$ | $0.66 \pm 0.07$ | $1.9 \pm 0.29$ | $0.339 \pm 0.005$ | $0.42 \pm 0.05$ | $0.109 \pm 0.01$ |
| Neural ODE (RK4) | $9.3 \pm 2.6$ | $0.73 \pm 0.07$ | $2.0 \pm 0.7$ | $0.32 \pm 0.04$ | $0.57 \pm 0.29$ | $0.12 \pm 0.04$ |
| Neural ODE (Tsit5) | $9.7 \pm 1.3$ | $0.724 \pm 0.03$ | $1.53 \pm 0.29$ | $0.26 \pm 0.06$ | $0.30 \pm 0.11$ | $0.085 \pm 0.019$ |
| cPLRNN | $5.3 \pm 1.9$ | $0.62 \pm 0.18$ | $1.6 \pm 0.5$ | $0.30 \pm 0.04$ | $0.37 \pm 0.14$ | $0.116 \pm 0.012$ |
| standard PLRNN | $3.84 \pm 0.23$ | $0.31 \pm 0.07$ | $1.18 \pm 0.26$ | $0.14 \pm 0.05$ | $0.24 \pm 0.06$ | $0.079 \pm 0.01$ |

