# OpenReview forum: "Continuous-Time Piecewise-Linear Recurrent Neural Networks"
_ICML.cc/2026/Conference — ICML 2026 regular_

### Official Review · Reviewer_UCEK · 2026-03-07

**Soundness:** 3
**Presentation:** 3
**Significance:** 3
**Originality:** 3
**Overall Recommendation:** 3
**Confidence:** 4

**Summary:**

The authors introduce cPLRNNs, a novel and interpretable method for learning from irregularly sampled time-series data. Training is made tractable through a custom interval root-finding algorithm and efficient by exploiting analytic solutions of linear systems within each subregion of the state space. Given a trained model, the framework can locate fixed points and limit cycles in the learned dynamical system. The method is evaluated on dynamical system reconstruction using synthetic Lorenz and LIF systems, as well as a real-world electrophysiology dataset.

**Compliance With Llm Reviewing Policy:**

Affirmed.

**Final Justification:**

My main concern is that the experiments presented and the claims made in the paper are not aligned. In the original review, I noted that the experiments did not explore irregularly sampled time series, which was the primary motivation of the work. While the authors provided experiments on irregularly sampled data during the rebuttal, these results raised further concerns regarding the rigor of the evaluation protocols, where the authors admit that they did not carefully control for important confounding factors in their reported metrics.

Additionally, the rebuttal results raised technical concerns about the performance of cPLRNNs in an irregularly sampled regime. Specifically, increasing the amount of data appears to lead to worse performance in the presented experiments (Table R5). This is highly counterintuitive and seems to undermine the effectiveness of the method. I would expect a method designed for irregular sampling to improve with more data rather than degrade. Because of these unresolved issues, I maintain my score.

**Key Questions For Authors:**

1. What does “interpretability” (line 21, Col 2) mean in this context? Can you design experiments that demonstrate that your method achieves your definition of interpretability?
2. Can you provide an example where cPLRNN outperforms standard PLRNN on a real-world dataset with irregularly sampled time series? The current electrophysiology experiment does not appear to evaluate cPLRNN under irregular sampling conditions. Since handling irregularly sampled data is a primary motivation of the work, the evaluation should more directly demonstrate this advantage.
3. Can you extend the synthetic dynamical systems analysis to more than 1 system to demonstrate that fixed point and limit cycle detection is reliable? Can you also evaluate if the proposed method is robust to the presence of process noise (noise over time)?
4. What kinds of insights do you gain by recovering the fixed points and limit cycles in the ephys example?
5. In table 2, why does the runtime of Neural ODE (Euler) decrease as P increases, while cPLRNN shows the opposite trend? If you keep increasing P, does cPLRNN ever become less efficient than Neural ODE (Euler)?

**Limitations:**

Yes, training can be unstable.

**Strengths And Weaknesses:**

Strengths:
- cPLRNN seems to be an well-motivted and novel new learning framework for irregularly sampled time-series.
- The derivation of analytic solution is well-motivated and a nice contribution to this framework.
- There are several important technical contributions that make learning tractable, such as the custom differentiable root finding algorithms.
- Well written and easy to understand.

Weakness:

- For all the tables, it’s unclear which values are the best. Is higher or lower score better? What does * mean? Please clarify and make it obvious, perhaps by bolding the best values!
- The authors mention “interpretability” as a key desiderata of a DS model, but do not define what it means and also do not have experiments aimed at validating that the model is interpretable.
- The evaluations primarily focus on regularly sampled time series, which does not align with the paper’s stated motivation of addressing irregularly sampled data. Consequently, it remains unclear from the presented experiments why cPLRNN would be preferable to standard PLRNN or NeuralODE (Euler).
- The fixed point detection and limit cycle algorithm are evaluated qualitatively, and not quantitatively. Therefore it’s unclear if cPLRNN improves over NeuralODE approaches or standard PLRNN. For instance, it is unclear which method is better in Figure 1. Can you quantify the accuracy of the fixed points/limit cycle detection? When does it fail?
- “Equation (14) holds for any type of limit cycle (stable, unstable, or saddle).” Is there a formal justification or proof for this claim? Additionally, how robust is this definition to noise? For example, noise in the sequence of subregions could perturb the observed transitions, potentially obscuring the detection of noisy limit cycles under this analysis.
- It’s unclear to me where the ${\bf x}_n$ is in the model formulation and what is being predicted. What loss function is optimized during training?

---

> ### Author Rebuttal · Authors · 2026-03-30
>
> We thank the referee very much for the constructive feedback!
>
> Rebuttal materials available here: https://drive.google.com/file/d/1oC-WJg_S36AqPBtz9oxBQxgTw8R_Q-p9/preview
>
> **Weaknesses**
>
> **W1 (tables unclear):** We now implemented all suggested changes in revised Tab. R1-R3.
>
> **W2 (interpretability):** With interpretability in this context of DS reconstruction the *mathematical tractability* of a model is meant (e.g. Eisenmann et al. 2023, 2026). In scientific or medical contexts, we would like to be able to analyze the math. properties of trained DS models further and thereby gain mechanistic insight into the underlying system dynamics. This is exactly what sect. 3.2 in our paper is about, with application of these techniques illustrated in Fig. 1 (location of fixed points), and for location of limit cycles in Fig. 2A,B (bottom).
>
> **W3 (focus on regular sampling):** We agree, and *provide in new Tabs. R4,R5 new experiments with irregularly sampled time series*, both synthetic and one real-world (PhysioNet) example. It further shows that performance of the discrete PLRNN, in contrast to the cPLRNN, degrades dramatically with less sampling times. Continuous-time models further come with *universal approximation guarantees* that discrete RNNs currently lack, see https://arxiv.org/abs/2602.08640, and their continuous nature also eases math. analysis of many geometrical properties (sect. 3.2).
>
> **W4 (quantification of fixed point/ limit cycle detection):** There might be a misunderstanding here that we will need to clarify: The techniques in sect. 3.2 for detecting fixed points and limit cycles in *trained* cPLRNNs provide *exact* solutions, there is no error! How well the different models recover the underlying system dynamics compared to each other is another question, which is answered by the $D_{stsp}$ & $D_H$ comparisons in Tabs. 1,3,4. The tools in sect. 3.2 are, instead, methods for *analyzing* geometrical structure in the state spaces of *trained models*. Numerically, it may happen that not all fixed points and limit cycles are detected, which is, however, true for *all* algorithms in this area which are usually numerical (in contrast to our semi-analytical technique).
>
> **W5 (claim about types of limit cycle):** Again, there might be a misunderstanding: A limit cycle of a DS is mathematically defined as an isolated closed orbit in state space. A solution to eqn. 14 will be *exactly* such a closed orbit, *by definition*. Stability in this context means whether there is convergence to or divergence from this limit cycle along one or more directions (as can be determined from the Jacobian eigenvalue spectra), but eqn. 14 is completely agnostic to this condition, it simply provides the exact solution for a closed orbit. Further, FPs & LCs are defined in the deterministic limit: While in practice noise may perturb the system’s state away from a cycle, for determining the limit cycle itself this does not matter.
>
> **W6 (what is the loss):** The $\mathbf{x}_n$ are the actual observed time series values which are to be predicted, and the loss is simply the MSE between these and predicted ($\tilde{\mathbf{x}}_n$) values. The $\mathbf{x}_n$ are generally coupled to the underlying latent DS model by a decoder model $\mathbf{x}_n=G(\mathbf{z}_n)$, see 1st pg. of sect. 3.1. We will add a sect. to the Appx. with all training details.
>
>  **Questions**
>
> **Q1 (interpretability):** Please see reply to W2 above.
>
> **Q2 (standard PLRNN with irreg. sampling):** See W3 above & new Tab. R4.
>
> **Q3 (add. example & process noise):** We have now produced *another synthetic example with process noise (new Tab. R5)*, on which we also show the *detection of a limit cycle, new Fig. R6*.
>
> **Q4 (insights gained by FP/LC analysis):** Generally, fixed points and limit cycles inform us about the type of system dynamics underlying the observed system’s behavior. In this particular case it tells us that the system is *bistable*, a stable fixed point and a stable cycle coexist and hence noise could push the system from being silent to spiking and vice versa (bistability is considered an important active memory mechanism in neuroscience, e.g. https://www.nature.com/articles/nn0898_273). We should have explained this and will add these insights in the revision.
>
> **Q5 (runtime as a func. of $P$):** The apparent decrease is merely due to fluctuations in memory/server load, as now evaluated more systematically *in new Fig. R5*. Formally, the Euler method should have a rather *fixed* runtime since it proceeds in fixed step sizes. Thus, for large $P$ Euler might become faster, but note it is included here merely to provide a kind of *lower runtime bound*, it is actually *not suitable for most nonlinear systems*: As indicated in sect. 4.3 & 4.4 and evident from its *much worse performance* (Tab. 1), for most systems it actually *diverges*, a well known issue in the numerical solver lit. (e.g. Press et al. 2007, https://numerical.recipes/book.html).

---

> > ### Author Rebuttal · Reviewer_UCEK · 2026-04-01
> >
> > I appreciate the effort for the new tables via Google Drive link. However, I will not evaluate it since **it's content far exceeds the 5000-character limit**, making it unfair to other papers that the make efforts to stay within the constraints.  Moreover, its unclear how these tables would fit into the main narrative since they are isolated from the submitted text. This points to a larger issue, that **the raised concerns require major revisions to be adequately addressed**, and **these should be handled on resubmission**. Therefore I maintain my score.

---

> > > ### Author Response · Authors · 2026-04-01
> > >
> > > Dear Referee,
> > >
> > > First, according to the ICML author guidelines, authors may provide an anonymous link to a document with additional figures and tables *in addition* to the 5000‑char limit. This option was equally available to all submissions, and our use of the link fully complies with these rules.
> > >
> > > Furthermore, the additional experiments suggested by the referee (W3) are an extension of those already presented in the paper to new irregularly sampled datasets. They therefore naturally align with, and further support, the existing narrative of our manuscript. This is also apparent from the fact that we could incorporate most new results into the already existing tables, which we had just reproduced in full (together with the previous results) in the provided document to put the new numbers directly in context.
> > >
> > > **We have now reproduced the specific results the referee asked for in W3 and Q5 below directly as tables.**
> > >
> > > Finally, we would like to note that five out of the six brought up weaknesses (all except W3) appear to be based on misinterpretations that we had hoped could clarify *within* the limits of the rebuttal window. Regarding W1 we also would like to point out that the information the referee asked for was already in the original table legends, although we agree should have been made clearer as we did now in the tables below (for all measures reported, lower values are better; the * was meant to indicate that many runs diverged – this is mentioned in the legend and main text, but unfortunately we forgot to add the * symbol).
> > >
> > > **Remarks on FINAL JUSTIFICATION:** We would like to briefly comment on two remarks by this referee in the final justification:
> > >
> > > > "... further concerns regarding the rigor of the evaluation protocols, where the authors admit that they did not carefully control for important confounding factors in their reported metrics."
> > >
> > > We are not sure what the referee is referring to here; our evaluation controls for all relevant variables, and there are no confounding factors in the reported metrics.
> > >
> > > > "Specifically, increasing the amount of data appears to lead to worse performance in the presented experiments (Table R5). "
> > >
> > > These are just chance variations, the differences here are not significant (Mann-Whitney U-test; $D_{stsp}$: U=15, 2-sided p=0.2544, $D_H$: U=20, 2-sided p=0.5941).
> > >
> > > We thank the referee for their time and efforts in evaluating our work, and hope that we could now clarify the referee’s remaining points.
> > >
> > > **Table R4:**
> > > Performance comparison on *irregularly sampled heartbeat time series* from the [PhysioNet/Computing in Cardiology Challenge 2012](https://physionet.org/content/challenge-2012/1.0.0/). The subject with the most datapoints was chosen and the time series delay-embedded ($\tau=1$ hour, $d=3$), normalized and split into a training (80%) and test trajectory (20%), with MSE evaluated on the test set.
> > > All values reported are across 5 runs (where for the standard PLRNN linear interpolation to full hours was used, see sect. 4). *Note that Euler diverges, causing huge errors. Bold = best.
> > >
> > >   Measure         | Neural ODE (Euler) | Neural ODE (Tsit5) | cPLRNN         | standard PLRNN |
> > >  |-----------------|--------------------|-----------------|----------------|----------------|
> > >  | MSE (test) $\downarrow$ | 4e3 +/- 6e3*      | 1.8 +/- 0.3       | **1.512 +/- 0.018** | 2.96 +/- 0.15  |
> > >
> > >
> > > **Table R5:**
> > > Performance of cPLRNN vs. standard PLRNN for variation of the percentage of available sampling points for the LIF example under *irregular* sampling, and performance comparison on a new synthetic example, the Izhikevich (2003, IEEE TNN) neuron model *with process noise* and *irregular* sampling from the cyclic regime. Values for $D_\text{stsp}$ and $D_H$ are given as median $\pm$ MAD. Bold = best.
> > >
> > >  | Model | $D_{stsp} \downarrow$ 10% LIF | $D_{stsp}\downarrow$  20% LIF | $D_{stsp}\downarrow$ Izhikevich | $D_H\downarrow$ 10% LIF | $D_H\downarrow$ 20% LIF | $D_H \downarrow$Izhikevich |
> > > |-------|----------------|----------------|-------------------|-------------|-------------|----------------|
> > > | cPLRNN | **0.26 +/- 0.03** | **0.33 +/- 0.25** | **7.8 +/- 0.3** | **0.232 +/- 0.024** | 0.26 +/- 0.5 | **0.746 +/- 0.003** |
> > > | standard PLRNN | 4.3 +/- 0.3 | 0.7 +/- 0.3 | 8.6 +/- 2.7 | 0.5 +/- 0.3 | **0.15 +/- 0.9** | 1 |
> > >
> > > **Figure R5 as Table:**
> > > Mean runtimes (error bar = standard deviation) for forward Euler integration of the Neural ODE ($M = 40$ fixed). All runs were started at the same time to achieve comparable server load. The apparent absence of any scaling with $P$ was statistically confirmed ($p=0.2499$, n.s.).
> > >
> > >   Model  | P = 2          | P = 5          | P = 10         | P = 20         | P = 30         |
> > >  |---------|----------------|----------------|----------------|----------------|----------------|
> > >  | Euler   | 55.0 ± 2.4 | 53.4 ± 2.0 | 53.7 ± 2.1 | 53.7 ± 2.1 | 53.1 ± 2.0 |

---

### Official Review · Reviewer_bjgg · 2026-03-11

**Soundness:** 4
**Presentation:** 4
**Significance:** 4
**Originality:** 3
**Overall Recommendation:** 5
**Confidence:** 4

**Summary:**

The paper introduces a continuous-time variant to PLRNNs (cPLRNN) intended for latent dynamical system (DS) modeling. When modeling non-linear DS, the piecewise-linear approach transfers the complexity of a non-linear evolution function to a partioning of its phase space into subregions hosting each a linear DS. This partitioning is naturally induced by ReLU activations within the model.
The main motivation behind this locally linear model is what the authors call here mathematical insight, that is access to methods from linear DS theory to analyze the resulting system and its properties (e.g. equilibria and limit cycles), which is usually not the case when resorting to non-linear "black-box" modeling. As was the case for the Neural ODE, introducing a continuous alternative to such models allows both a better proximity to real-life use cases (often in continuous time), and robustness to data irregularly spaced in time (which RNNs are known to be inefficient to). The locally linear nature of the model also enables analytic formulation of its (local) solution(s), bypassing the need to a computationnally expensive numerical integration. The main difficulty arising from this structure is gradient computation taking "switching times" into account, time indices where a trajectory enters a new subregion and change its behaviour. This requires a specific root-finding algorithm overriding automatic differentiation. The architecture is tested on two synthetic datasets (the chaotic Lorenz attractor and discontinuous LIF model) and a real-world dataset (electrophysiological single neuron recordings) and compared to Neural ODEs with different integration schemes and the standard PLRNN model. The overall outcome is that the cPLRNN performs similarly to a neural ODE (with a well-chosen scheme) metric-wise but is significantly cheaper to compute, while retaining its analytical advantage. The cPLRNN also tends to be a bit less effective than a standard PLRNN of regularly sampled data but performs better in irregularly sampled contexts, as it would be expected.

**Compliance With Llm Reviewing Policy:**

Affirmed.

**Final Justification:**

The authors addressed my concerns and questions in the rebuttal. It reinforced my prior assessment to accept the paper, which is motivated by its soundness (S1, S2), its clarity (S3) and its originality (S4).

**Key Questions For Authors:**

1. How often does the ill-conditioned eigen-decomposition issue occurs ? Is the occurence rate linked to the latent dimension or the amount of ReLU activations ?

2. The encoder/decoder structure used for experiments is the rather trivial identity observer. What are your thoughts on the link between the latent DS and the potentially unknown systems generating data and their respective properties, when using a more complex structure such as an MLP ?

**Limitations:**

yes

**Strengths And Weaknesses:**

**Strengths**

1- The paper is technically sound. Its theoretical aspects are built upon the PLRNN and developped using standard concepts from dynamical system theory.

2- The main technical contribution would be the algorithmic work towards finding switching times, which is elegantly brought up in the paper and has a lot of details available in the appendix. The experimental tryouts are well designed, with adequate metrics and use-cases displaying specific properties. Benchmarking solely against both "parent" architectures (PLRNN and neural ODE) might give a narrow view of the performances, but they are the only architectures immediately comparable with the cPLRNN.

3- The paper well structured and easy to read, even while I was missing beforehand knowledge about the PLRNN. I personally believe that, in the field of dynamical system reconstruction and more generally in ML applied to physical systems, the main driving factor should be abandonment of black-box modeling. This paper brings a novel architecture whose principal quality is its easily analyzable structure and is thus, in my mind, completely relevant.

4- The cPLRNN naturally inscribes itself in the wake of the PLRNN and the neural ODE. While not a totally new approach, it is a novel architecture and adds valuable new insights.

**Weaknessess**

1- The authors mention a numerical robustness issue concerning ill-conditioned eigen-decomposition which was still being investigated.

2- The cPLRNN's only structural hyperparameters are the latent dimension and amount of ReLU activations. It seems logical that the easily analyzable structure of the single-layer cPLRNN comes as a tradeoff with the expressivity of deeper black-box models. While not stricly speaking a limitation, I think it could have been briefly discussed.

---

> ### Author Rebuttal · Authors · 2026-03-30
>
> We thank the referee very much for the very supportive feedback and for raising a couple of interesting questions!
>
> Additional rebuttal materials can be found here: https://drive.google.com/file/d/1oC-WJg_S36AqPBtz9oxBQxgTw8R_Q-p9/preview
>
> **Weaknesses**
>
> **W1 (ill-conditioned matrices):** For this we have an ad-hoc solution (perturbing the matrices slightly if this problem is encountered, see Appx. C.4). Although not completely satisfying, it addresses the problem reasonably well. We now also checked how often this problem actually occurs during training, and found that only a marginal percentage (<$10^{-5}$%) of all Jacobians is degenerate, cf. *new Fig. R5*. Therefore our perturbation method should introduce only a negligible error.
>
> **W2 (tradeoff analytical tractability vs. expressivity):** Yes, we agree, and will discuss this in more detail in the final sect. 5. Strictly speaking, mathematically the same methodology could be applied to deeper networks, but the resulting combinatorial increase in the number of linear subregions would still impede analysis. In the light of recent observations, however, that actually with only a few linear subregions attractor geometries (Brenner et al. 2024, *NeurIPS*) as well as behavior on different computational tasks (https://openreview.net/forum?id=qI2Vt9P9rl) can be well captured, keeping the network structure parsimonious is a reasonable choice.
>
> **Questions**
>
> **Q1 (frequency of occurrence of ill-conditioned matrices):** See our reply to W1: We analyzed this now, and ill-conditioned matrices are a rather rare event, such that our perturbation technique should only introduce negligible error. How often this occurs also does not depend too strongly on the number of linear subregions, *see new Fig. R5*, and in any case the percentage remains extremely low.
>
> **Q2 (encoder-decoder structure):** This is a great question – discrete-time PLRNNs have indeed been embedded into much more complex encoder-decoder structures (e.g. https://proceedings.mlr.press/v235/brenner24a.html), although one observation in this lit. is that making the encoders and decoders too expressive actually removes too much of the burden from the latent DSR model to reconstruct the underlying dynamics, so there seems to be a tradeoff. Either way, since an important part of this paper is direct computational and performance  comparisons between our solution algorithms for the latent dynamics and those introduced with Neural ODEs, we intentionally wanted to keep the formulation of the encoders and decoders simple to avoid any computational overhead and focus the comparisons on the key aspects.

---

> > ### Author Rebuttal · Reviewer_bjgg · 2026-04-02
> >
> > Thank you for addressing my comments. I keep my recommendation to accept.

---

> > > ### Author Response · Authors · 2026-04-02
> > >
> > > We once again thank the referee for the careful and supportive evaluation of our manuscript. We will make sure to integrate all our replies to the points raised above into a final version.

---

### Official Review · Reviewer_S8Cj · 2026-03-13

**Soundness:** 3
**Presentation:** 3
**Significance:** 2
**Originality:** 2
**Overall Recommendation:** 4
**Confidence:** 3

**Summary:**

This paper proposes a continuous-time version of PLRNNs (cPLRNN) for dynamical system reconstruction. The main idea is to exploit the piecewise-linear structure to derive semi-analytic trajectories within regions, identify switching times by root finding, and then use the resulting model for dynamical analysis such as fixed points and limit cycles. The paper also claims advantages for irregularly spaced observations.

**Compliance With Llm Reviewing Policy:**

Affirmed.

**Final Justification:**

The authors have provided additional baselines and addressed my main concerns regarding scalability. Thus, I am raising my score to 4. However, I have remaining questions about the hyperparameter tuning for the baselines, which I have posted in the acknowledgment.

**Key Questions For Authors:**

1. Can the authors better clarify the novelty relative to prior continuous-time and piecewise-linear neural dynamical models?
2. How does runtime scale with the number of switching events / visited regions?
3. How sensitive is the limit-cycle solver to initialization, and what is its practical computational cost?
4. Why were stronger baselines for irregularly sampled time series not included?
5. Under what conditions is the continuous approximation appropriate for discontinuous systems such as LIF dynamics?

**Limitations:**

yes

**Strengths And Weaknesses:**

Strengths:
1. The problem is relevant: continuous-time, interpretable latent dynamics with support for irregular sampling is an important topic.
2. The use of the piecewise-linear structure for semi-analytic simulation is interesting and more tailored than treating the model as a generic Neural ODE.
3. The paper goes beyond prediction and attempts post-hoc dynamical analysis (fixed points / limit cycles), which is valuable for scientific applications.
4. Experiments suggest that the method can be competitive while being faster than some standard Neural ODE solvers.

Weaknesses:
1. Modeling novelty appears limited. The continuous-time cPLRNN is obtained mainly by rewriting the discrete-time PLRNN update as an ODE while keeping essentially the same ReLU / piecewise-linear right-hand side. Therefore, the main contribution seems to lie less in a new continuous-time model class and more in the semi-analytic solver, switching-time root finding, and downstream dynamical analysis. The paper should clarify more precisely what is genuinely new at the modeling level versus at the algorithmic/analysis level.
2. Continuous-time neural dynamical models are not new. The main novelty seems to be the cPLRNN-specific semi-analytic solver and analysis pipeline, rather than simply using a ReLU network in an ODE. The paper should clarify more precisely what is new relative to prior Neural ODE, Latent ODE, NCDE, and other piecewise-linear dynamical models.
3. Scalability and runtime claims are not fully convincing.
The state space is partitioned into many linear regions, and in practice the number of switching events may make simulation and analysis costly. The paper acknowledges that performance degrades when many switching boundaries are present, so stronger scaling evidence would be helpful. I am also not fully convinced by the runtime comparison as presented.
4. The limit-cycle analysis seems potentially expensive and initialization-sensitive.
The fixed-point analysis is straightforward, but the practical value of the limit-cycle part is less clear. Solving Eq. (14) appears potentially costly and depends on guessed region sequences / flight times, yet there is no complexity or robustness analysis.
5. Irregularly spaced observations are addressed conceptually, but empirical validation is limited.
The paper explains how arbitrary observation times are handled by evaluating the continuous-time piecewise solution directly at those times. However, the empirical support is still limited, and stronger irregular-time baselines (e.g., Latent ODE / NCDE style models) are missing.
6. Modeling discontinuous systems with a continuous surrogate is not sufficiently discussed.
In Section 4.3, a continuous piecewise-linear model is used to fit a system with hard threshold/reset discontinuities. The empirical result is interesting, but the paper does not discuss when this approximation is theoretically justified or what its limitations are.
7. Sparse teacher forcing needs clearer explanation.
It should be stated more explicitly in the main text how sparse teacher forcing is applied during training versus testing, and whether results are sensitive to the forcing interval.

Minor comments:
1. There are several writing / formatting issues (e.g., placeholder-like references such as “Authors, 2026”, grammar/typo issues, and table presentation).
2. The experimental examples are somewhat simple and mostly low-dimensional.

---

> ### Author Rebuttal · Authors · 2026-03-30
>
> We thank the referee for the fair comments and identifying issues that need further attention.
>
> Add. rebuttal materials available here: https://drive.google.com/file/d/1oC-WJg_S36AqPBtz9oxBQxgTw8R_Q-p9/preview
>
> **Weaknesses**
>
> **W1 (novelty):** The major contribution of the Neural-ODE paper (2018) was in our minds not so much contin. RNNs/DNNs (which basically already existed since the 70s), but an efficient algo for the loss which did not require back-prop through the numerical solver. We see our contrib. similarly: Rewriting the discrete as cPLRNN is not a major leap, as the ref. correctly points out, but we introduce a completely novel type of solver operating on a very different principle (determining switching times between boundaries) which facilitates both optimization & model simulation. In addition, we provide a set of analysis tools exploiting the same properties. This is very novel, in our minds.
>
> **W2 (contributions):** Yes, see W1, we do not claim that either providing a contin. time or a PL model is in itself novel. The novelty is clearly the type of solution algo for both optim. & simulation, and the analysis package that comes with it.
>
> **W3 (scalability):** Fair point. We have performed a more extensive analysis of how model runtimes scale as a func. of the number of linear subregions $2^P$, the number of latent states $M$, and the number of linear regions visited (switches); *results in new Fig. R1* show that our algo scales linear or even sublinear in all cases. We further would like to emphasize that recent studies showed that often only a fairly low number of linear subregions is required to correctly capture an attractor’s geometry (Brenner et al. 2024, *NeurIPS*) or to perform a variety of cognitive tasks (https://openreview.net/forum?id=qI2Vt9P9rl), suggesting that in many practical scenarios $P$ does not have to be large.
>
> **W4 (comp. complexity of limit cycle analysis):** As the analysis in *new Fig. R2* shows, limit cycles are generally found very fast (<20 sec on a 8-core CPU), and detection is robust against misspecification of boundary crossing points and flight times. The detection of limit cycles with “traditional” numerical continuation methods, instead, requires full numerical integration of the differential equations, unstable cycles are tricky to find, and half-stable cycles cannot be detected at all. Compared to these standards in the field, we think that our root finding procedure is actually quite efficient. *Most importantly*, note that the analysis is only required *once* posthoc, after a good model has been found, such that the fact that our algo is *exact* is actually the more important aspect.
>
> **W5 (irregular sampling & stronger baselines):** We agree and now provide *several new analyses in new Tabs. R4 & R5* with one more simulated and one empirical (PhysioNet) example, plus a comparison showing that the discrete PLRNN substantially degrades with data sparsity while the cPLRNN does not. We further updated Tab. 1, 3, & 4 with *new contin.-time baselines (Tab. R1-R3), namely SINDy, Neural CDE, & Latent ODE*, as suggested. In our hands, Neural CDE & Latent ODE actually performed quite poorly using the provided default code, while SINDy only performed good on the one example where its polynomial library matched exactly the ground truth (Lorenz-63), but completely broke down for the other examples where this was not the case, as reported previously (Hess et al. 2023).
>
> **W6 (systems with hard thresholds):** There may have been a misunderstanding here: note that *both* the cPLRNN and LIF model are *continuous in time*, but *both* are *discontinuous* in either Jacobian’s across subregion boundaries or states. We had included the LIF in fact as an example of a system which is presumably well captured by a PL description. We indeed observed that switching times often aligned with the spike times of the LIF, *as shown in new Fig. R4*.
>
> **W7 (sparse teacher forcing):*** STF is a technique introduced in Mikhaeil et al. (2022, *NeurIPS*) to avoid exploding gradients when training on chaotic systems. It is only used in *training*; during test time the model is just left to freely evolve without any forcing. As shown in Mikhaeil et al., STF indeed depends on the forcing interval, with an optimum achieved when it corresponds to the Lyapunov predictability time of the underlying DS. In practice, the optimal forcing interval is often just determined by grid search. We will add a sect. to the Appx. with formal details on STF.
>
> **Minor points**
>
> **M1)** We will check the ms. carefully before it moves further (note that "Authors"-links were required qua ICML policy for parallel subm.).
>
> **M2)** See W5, we have added another complex real-world example.
>
> **Questions**
>
> **Q1 (novelty):** Please see W1 & W2.
>
> **Q2 (scaling):** Please see reply to W3.
>
> **Q3 (limit cycle solver):** Please see W4.
>
> **Q4 (baselines):** Please see reply to W5.
>
> **Q5 (“contin. approx.’’):** See W6 above.

---

> > ### Author Rebuttal · Reviewer_S8Cj · 2026-04-03
> >
> > I appreciate the authors' extensive efforts during the rebuttal phase. They have addressed my main concerns by providing additional experiments on a real-world dataset (PhysioNet) and comparing against the suggested baselines (Latent ODE, NCDE, SINDy). They also clarified the scalability and the computational cost of their limit-cycle analysis.
> >
> > Therefore, I am raising my score to 4.
> >
> > However, I do have two remaining reservations that prevent a higher score:
> >
> > 1. The authors noted that the baselines (like Latent ODE/NCDE) performed poorly using 'default code'. Given how notoriously sensitive these models are to hyperparameters, this comparison might not fully reflect their true capacity.
> >
> > 2. The authors circumvented the rebuttal text limits by placing extensive experimental details and discussions into exceptionally long table captions in the external PDF. While technically not violating the anonymity policy, this practice borders on unfairness to other authors who strictly adhered to the limits.

---

> > > ### Author Response · Authors · 2026-04-07
> > >
> > > We very much thank the referee for the appreciation of our additional results and the time taken to carefully and constructively evaluate both our original submission and our rebuttal.
> > >
> > > We would like to take this final opportunity to comment on the referee’s two remaining points:
> > >
> > > 1) We have started a much more extensive grid search, varying for Latent-ODE the learning rate, latent dimension, recognition ODE state width, ODE MLP hidden width, GRU network width, the depths of the recognition and generative ODE MLPs, batch size, and observed timepoint subsampling rate, and for NCDE the learning rate, batch size, and sequence length, each across 2–4 levels. Among the 199/272 runs for Latent-ODE and 173/522 runs for NCDE that finished so far, none even came close to the performance levels of the cPLRNN (see tables below). This appears to be rooted in the fact that both these models are simply not well suited for the type of DS reconstruction tasks that are the focus of our paper: They do not easily allow for generating longer autonomous rollouts, but most commonly *either diverge or run into fixed points*, causing their much worse performance on most of the tasks (which require limit cycle or chaotic behavior).
> > >
> > > 2) We see the point that some of the table legends may seem rather long, but please note that to a fair degree these just *repeat* descriptive content that is already present verbatim in the original paper (all the black parts in Tab. R1-R3). According to our understanding of the ICML 2026 author guidelines, the figures & tables with captions in the external document do not count toward the 5000‑char rebuttal limit. Our intention was to collect all the new results with context in a single place, rather than distributing them across rebuttals.
> > >
> > > We hope this makes sense, and thank the referee once again for engaging so deeply with our submission and for raising their score.
> > >
> > > **Table R1b.**
> > > Comparison of reconstruction performance between cPLRNN, Neural CDE, and Latent ODE on the Lorenz-63 dataset. Reported are geometrical ($D_{\mathrm{stsp}}$) and temporal ($D_\text{H}$) disagreement (lower is better) as median $\pm$ MAD.  \**Best* value across *all* runs finished.
> > >
> > >   Model       | $D_{\mathrm{stsp}}$$\downarrow$ | $D_\text{H}$$\downarrow$ |
> > >  |-------------|----------------------------------|--------------------------|
> > >  | cPLRNN      |  **0.35 $\pm$ 0.14**                    | **0.116 $\pm$ 0.012**          |
> > >  | Neural CDE  | $11.2\pm0.1$                     | $0.856\pm0.001$          |
> > >  | Latent ODE  | $9.78^*$    			|$0.871$       |
> > >
> > > **Table R2b.**
> > > Performance comparison for cPLRNN, Neural CDE, and Latent ODE trained on *irregularly* sampled data from the LIF model. Reported are median $\pm$ MAD for geometrical ($D_{\mathrm{stsp}}$) and temporal ($D_\text{H}$) disagreement (lower is better), and MAE for short-term prediction. \**Best* value across *all* runs finished.
> > >
> > >   System               | $D_\text{stsp}$$\downarrow$ | $D_\text{H}$$\downarrow$ | MAE$\downarrow$ |
> > >  |----------------------|-----------------------------|---------------------------|-----------------|
> > >  | **Irregularly sampled**|                          |                           |                 |
> > >  | cPLRNN               | **0.26 $\pm$ 0.03**   | **0.232 $\pm$ 0.024** | **0.064 $\pm$ 0.014** |
> > >  | Neural CDE           | $6.9^*$              | $0.86$        | $0.43$ |
> > >  | Latent ODE           | $13.8^*$              | $0.85$         | $0.43$  |

---

### Official Review · Reviewer_LJYU · 2026-03-13

**Soundness:** 3
**Presentation:** 2
**Significance:** 3
**Originality:** 3
**Overall Recommendation:** 5
**Confidence:** 3

**Summary:**

The paper presents a continuous Piecewise Linear Recurrent Neural Networks (cPLRNN) for the reconstruction of dynamical systems (DSR).
Classical PLRNNs work well for DSR in practice, and they are interpretable thanks to their piecewise linear regions, but they cannot really model continuous time DS, since they are defined as discrete-time updates. Neural ODEs (NODEs) solve the issue of continuous time modeling, but they are slow and often unstable to train.

This work aims to join the strengths of NODEs and PLRNNs, by proposing a continuous time PLRNN. Leveraging the piecewise linear structure of the dynamics, they manage to discretize and train the network without actually perform numerical integration which is typically required in neural ODEs. The proposed approach involves instead solving analytically the differential equation inside each of the linear regions and searching for switching times between regions. The derivative at switching times is approximated through the implicit function theorem.

Experiments were conducted on reconstruction of 3 dynamical systems: Lorenz-63, a Leaky Integrate-and-Fire model  of a neuron (with regularly and irregularly sampled points) and a recording of membrane potential . Overall cPLRNN behaves similarly to its discrete counterpart, and similarly or better than Neural ODE. On the LIF model with irregularly sampled points, cPLRNN and the neural ODE models provide good reconstruction, while the discrete RNN struggles significantly.

**Compliance With Llm Reviewing Policy:**

Affirmed.

**Final Justification:**

I still think the paper’s impact is somewhat specialized to the dynamical systems reconstruction community, and the practical advantage over discrete PLRNNs on regularly sampled data remains limited. However, the contribution is technically solid, the methodology is original, and the rebuttal resolves my main concerns.

I am therefore raising my score.

**Key Questions For Authors:**

1) Can you elaborate more on the advantage and necessity of continuous-time modeling, either providing additional empirical validation on irregularly sample data, or showing classes of systems that discrete time models can not easily model?
2) How does the model complexity in time scale when the number of relu variables $P$ and the state size $M$ increase?
3) Can you include more baselines, as suggested in the weakness section?

**Limitations:**

The authors discussed the limitations of the model in a fair way, highlighting for example possible instability and suggesting ways to mitigate it.

**Strengths And Weaknesses:**

Overall the paper is well-positioned in the DSR community, offering a new model that gives good generation of trajectories of dynamical systems, while keeping it mathematically easy to analyze like previous works in the field. However, the significance of the work is restricted to the specific field, lacking broader baselines. Moreover, the advantage against classic PLRNNs is not completely evident from the empirical validation.

## Strenghts

1) The work is fairly well written and presented.  All the metrics, tasks and baselines are clearly defined.

2) The work is overall sound in the presentation. It provides clear and not too dense mathematical derivation of the motivation behind the design choices composing cPLRNN.

3) The main motivation of the work is that current SoTA for DSR is mainly based on discrete systems. This is a reasonable concern, since most real dynamical systems  are continuous in nature.
4) The work presents a novel method to train the model. By employing a  semi-analytical formulation, the model avoids numerical integration. This in turn makes training cPLRNN significantly faster than NODE approaches, which rely on backpropagating through integration steps.
## Weaknesses

1) While the problem presented, and solved by the model, is potentially relevant, the empirical advantage over discrete PLRNNs is not completely demonstrated in the current experiments. Overall, discrete PLRNNs behave generally equally good or better than cPLRNN, except on only one experiment performed with irregularly sampled dynamical systems. Furthering the experimentation on a broader set of irregularly sampled data, or giving examples of  real-world systems where discrete formulations fail, would strengthen the motivation of introducing a class of continuous time rnns, considering also the slower and more unstable training.
2) The number of baselines could be expanded, by including other sota models. It would be useful to compare cPLRNN with models such as Neural CDE (Kidger et al., 2020) on irregularly sampled dynamical systems, or with the Echo State network (Jaeger and Haas, 2004) on chaotic dynamical systems, or with other interpretable models such as Sparse identification of nonlinear dynamics (Brunton et al., 2016). These approaches would be beneficial to the significance of the paper, offering fair comparisons for the proposed models.
3) The scalability of the model is unclear: the approach relies on detecting switching times between linear regions, which may become increasingly complex as the number of ReLU units grows. . The model is shown to be efficient with maximum $P=10$, but a discussion of complexity and scalability w.r.t. the number of regions would be useful.

Furthermore, it would strengthen the evaluation to report prediction horizons in units of Lyapunov time for chaotic benchmarks (e.g., Lorenz), which would quantify how long trajectories remain accurate relative to the intrinsic predictability limit of the system. Does the continuous time formulation help in the limit?

## References

 Kidger, Patrick, James Morrill, James Foster, and Terry Lyons. ‘Neural Controlled Differential Equations for Irregular Time Series’. arXiv:2005.08926. Preprint, arXiv, 5 November 2020. [https://doi.org/10.48550/arXiv.2005.08926](https://doi.org/10.48550/arXiv.2005.08926).

Jaeger, Herbert, and Harald Haas. ‘Harnessing Nonlinearity: Predicting Chaotic Systems and Saving Energy in Wireless Communication’. _Science_ 304, no. 5667 (2004): 78–80. [https://doi.org/10.1126/science.1091277](https://doi.org/10.1126/science.1091277).

Brunton, Steven L., Joshua L. Proctor, and J. Nathan Kutz. ‘Discovering Governing Equations from Data: Sparse Identification of Nonlinear Dynamical Systems’. _Proceedings of the National Academy of Sciences_ 113, no. 15 (2016): 3932–37. [https://doi.org/10.1073/pnas.1517384113](https://doi.org/10.1073/pnas.1517384113).

---

> ### Author Rebuttal · Authors · 2026-03-30
>
> We thank the referee for the careful reading and supportive feedback on our manuscript!
>
> Additional rebuttal materials can be found here: https://drive.google.com/file/d/1oC-WJg_S36AqPBtz9oxBQxgTw8R_Q-p9/preview
>
> **Weaknesses**
>
> **W1 (advantage over discrete PLRNNs):** Yes this is a completely valid point. For regularly sampled time series we would usually not expect an advantage of a continuous formulation *performance-wise*. We stress, however, it would still be advantageous for *model analysis*, for *extrapolating to arbitrary time points*, and, as shown recently (https://arxiv.org/abs/2602.08640), continuous-time RNNs come with *universal approximation guarantees* for infinite time dynamical systems that discrete time RNNs currently lack. We now included two more experiments (one simulated, one empirical: PhysioNet/Computing in Cardiology Challenge) with irregular sampling that further demonstrate the advantage of the cPLRNN in this regime, *see new Tables R4 & R5 at the link provided above*. We further added to Tab. R5 an evaluation of how the discrete and contin. PLRNN behave with different amounts of sampling time points, and found that while the cPLRNN is only mildly affected as the mean and variance of the inter-sampling intervals are increased, the performance of the discrete PLRNN strongly degrades.
>
> **W2 (other baselines):** As suggested, we now added other continuous-time models for DSR, namely SINDy (Brunton et al. 2016), Neural CDEs (Kideger et al. 2020), and Latent ODE (Rubanova et al. 2019). *Results are in updated Tables R1-R3*. As observed previously (Hess et al. 2023, *ICML*), SINDy works best if the library functions exactly match the ground truth (GT) system, as is the case for the Lorenz-63 (as both SINDy and Lorenz-63 are low-order polynomials), but strongly degrades and falls far behind the cPLRNN in the more natural setting where the GT equations are not precisely known. Similarly, Neural CDE and Latent ODE perform far worse (although better results may be achieved with more extensive hyper-parameter search than was possible within the 1-week rebuttal period, the observed performance differences are actually so big that it seems unlikely results would completely reverse).
>
> **W3 (scalability):** Fair point. *We now performed a more extensive scaling analysis and show in new Fig. R1* how model runtimes scale as a function of the number of linear subregions ($2^P$), number of latent states $M$, and number of linear regions visited – in all cases the scaling is linear or even sublinear. Moreover, we would like to emphasize that a couple of recent studies have demonstrated that often only a fairly low number of linear subregions is required to correctly capture an attractor’s topology or geometry (Brenner et al. 2024, *NeurIPS*) or to perform a variety of cognitive tasks (https://openreview.net/forum?id=qI2Vt9P9rl), suggesting that in many practical scenarios $P$ does not have to be very large.
>
> **W4 (prediction horizons):** We have added *prediction times in units of Lyapunov times as a further metric to Table R1*, as suggested, but this does not alter the results.
>
> **Questions**
>
> **Q1 (advantages of continuous-time modeling):** See our reply to W1 above: We have added further empirical validation (*new Tabs. R4, R5*), but also remark that, besides the inability of discrete time models to efficiently deal with irregular sampling, they cannot inter- or extrapolate solutions to arbitrary time points, and, more importantly, the non-smooth nature of their state spaces also makes post-hoc analysis of trained models more challenging (e.g., solution curves or basin boundaries will in general not be continuous, differentiable sets). Discrete time systems can also not as easily approximate any arbitrary DS in the infinite time limit as continuous-time models, as has recently been established mathematically (https://arxiv.org/html/2602.08640v2).
>
> **Q2 (scalability):** See our response to W3 above and *new Fig. R1* where we investigated the scaling with $P$ (# ReLUs) and $M$ (state dimension).
>
> **Q3 (more baselines):** Yes, we now did this, see reply to W2 above, and revised Tables R1-R3.

---

> > ### Author Rebuttal · Reviewer_LJYU · 2026-04-03
> >
> > Thank you for the rebuttal and additional experiments.
> >
> > I still think the paper’s impact is somewhat specialized to the dynamical systems reconstruction community, and the practical advantage over discrete PLRNNs on regularly sampled data remains limited. However, the contribution is technically solid, the methodology is original, and the rebuttal resolves my main concerns.
> >
> > I am therefore raising my score.

---

> > > ### Author Response · Authors · 2026-04-04
> > >
> > > We are glad we were able to resolve the referee’s concerns and very much appreciate the thoughtful comments on the paper’s impact and target communities. While dynamical systems reconstruction is indeed a primary audience, we also see the methods as relevant for irregular time‑series forecasting (as in the PhysioNet example), and will clarify this more explicitly in the revision. We thank the referee again for the thorough engagement with our manuscript.

---

### Decision · Program_Chairs · 2026-04-30

**Decision:**

Accept (regular)

**Comment:**

The submission proposes Continuous-Time Piecewise-Linear Recurrent Neural Networks (cPLRNN), a framework designed for dynamical systems reconstruction. By extending discrete-time PLRNNs into a continuous formulation, the work aims to reconcile the high performance of PLRNNs with the continuous nature of physical and biological processes. The primary contribution is a semi-analytical algorithm for training and simulation that exploits the piecewise-linear structure of the model to bypass the need for computationally expensive numerical integration typically required by neural ODEs. Methods are also presented to determine fixed points and limit cycles within trained models. The framework is specifically positioned to handle irregularly sampled data and provide mathematical tractability for mechanistic insight, outperforming standard NODEs in efficiency and DSR performance on benchmarks like the Lorenz-63 and Leaky Integrate-and-Fire (LIF) models.

Reviewers highlighted the novel semi-analytical solver as a major strength. By calculating switching times between linear regions and solving the differential equations analytically within those regions, the model achieves significantly faster training than standard Neural ODE solvers that rely on backpropagating through integration steps. Reviewers also appreciated the model's ability to go beyond simple prediction.

The authors raised concerns regarding the criticisms raised by Reviewer UCEK. I agree with the author response that the increase in the cPLRNN error is likely a non-issue; the use of the Mann-Whitney U test seems appropriate to demonstrate that nothing suspicious need have happened here, just bad luck. Some misunderstandings regarding the ICML 2026 guidelines were cause for concern, but it appears that the reviewer did consult the additional findings as required of them. There are one concerns raised by Reviewer UCEK that do stand: testing on a single subject from the PhysioNet Challenge is fine for a quick rebuttal, but it is still a weak experiment that does not entirely succeed at an effective comparison on irregularly sampled time series.

Another concern raised by Reviewer S8Cj involves the use of default code for baseline comparisons to latent ODE/NCDE. This is always a difficulty when performing comparisons using older methods, especially those that are more sensitive to hyperparameters. The final rebuttal by the authors show results after an extensive grid search, which is excellent to see, and truly addresses this concern in my view.

All other concerns appear to have been addressed during the discussion period, and is suggestive of a submission that is ripe for publication. There are many results presented in the discussion that absolutely need to go into a final version, including the experiments with PhysioNet, ideally treated over a much more sizable proportion of the subjects. Conditional on these inclusions, I recommend acceptance.